# Recent improvements and maximum covariance analysis of aerosol and cloud properties in the EC-Earth3-AerChem model

**Manu Anna Thomas**[1], **Klaus Wyser**[1], **Shiyu Wang**[1], **Marios Chatziparaschos**[2], **Paraskevi Georgakaki**[4], **Montserrat Costa-Surós**[2], **Maria Gonçalves Ageitos**[2,3], **Maria Kanakidou**[6], **Carlos Pérez García-Pando**[2,7], **Athanasios Nenes**[4,8], **Twan van Noije**[5], **Philippe Le Sager**[5], and **Abhay Devasthale**[1]

[1]Swedish Meteorological and Hydrological Institute (SMHI), Folkborgsvägen 17, 60176 Norrköping, Sweden

[2]Barcelona Supercomputing Center (BSC), Plaça d'Eusebi Güell 1-3, Les Corts, 08034 Barcelona, Spain

[3]Department of Project and Construction Engineering, Universitat Politècnica de Catalunya (UPC), C. Jordi Girona 31, 08034 Barcelona, Spain

[4]Laboratory of Atmospheric Processes and their Impacts (LAPI), School of Architecture, Civil and Environmental Engineering, École Polytechnique Fédérale de Lausanne (EPFL), 1015 Lausanne, Switzerland

[5]The Royal Netherlands Meteorological Institute (KNMI), Utrechtseweg 297, 3731 De Bilt, the Netherlands

[6]Department of Chemistry, University of Crete, Voutes University Campus, Heraklion, Crete, Greece

[7]Catalan Institution for Research and Advanced Studies (ICREA), Passeig Lluís Companys 23, 08010 Barcelona, Spain

[8]Center for Studies of Air Quality and Climate Change, Institute of Chemical Engineering Sciences (ICE-HT), Foundation for Research and Technology Hellas (FORTH), 26504 Patras, Greece

**Correspondence:** Manu Anna Thomas (manu.thomas@smhi.se)

**Abstract.** Given the importance of aerosols and clouds and their interactions in the climate system, it is imperative that the global Earth system models accurately represent processes associated with them. This is an important prerequisite if we are to narrow the uncertainties in future climate projections. In practice, this means that continuous model evaluations and improvements grounded in observations are necessary. Numerous studies in the past few decades have shown both the usability and the limitations of utilizing satellite-based observations in understanding and evaluating aerosol–cloud interactions, particularly under varying meteorological and satellite sensor sensitivity paradigms. Furthermore, the vast range of spatio-temporal scales at which aerosol and cloud processes occur adds another dimension to the challenges faced when evaluating climate models.

In this context, the aim of this study is two-fold. (1) We evaluate the most recent, significant changes in the representation of aerosol and cloud processes implemented in the EC-Earth3-AerChem model in the framework of the EU project FORCeS compared with its previous CMIP6 version (Coupled Model Intercomparison Project Phase 6; https: //pcmdi.llnl.gov/CMIP6/, last access: 13 February 2019). We focus particularly on evaluating cloud physical properties and radiative effects, wherever possible, using a satellite simulator. We report on the overall improvements in the EC-Earth3-AerChem model. In particular, the strong warm bias chronically seen over the Southern Ocean is reduced significantly. (2) A statistical, maximum covariance analysis is carried out between aerosol optical depth (AOD) and cloud droplet (CD) effective radius based on the recent EC-Earth3-AerChem/FORCeS simulation to understand to what extent the Twomey effect can manifest itself in the larger spatio-temporal scales. We focus on the three oceanic low-level cloud regimes that are important due to their strong net cooling effect and where pollution outflow from the nearby continent is simultaneously pervasive. We report that the statistical covariability between AOD and CD effective radius is indeed dominantly visible even at the climate scale when the aerosol amount and composition are favourably preconditioned to allow for aerosol–cloud interactions. Despite this strong covariability, our analysis shows a strong cooling/warming in shortwave cloud radiative effects at the top of the atmosphere

in our study regions associated with an increase/decrease in CD effective radius. This cooling/warming can be attributed to the increase/decrease in low cloud fraction, in line with previous observational studies.

## 1 Introduction

Aerosols can potentially act as cloud condensation nuclei (CCN) or ice nuclei, influencing cloud formation, structure and properties. The type, concentration and size distribution of aerosols impact cloud microphysics, altering droplet or ice crystal size, number concentration, and cloud albedo (Twomey, 1974, 1977; Albrecht, 1989; Ramanathan et al., 2001). Conversely, clouds can also affect aerosols through wet deposition processes and also indirectly via their impact on meteorology and gas phase photochemistry, thereby influencing their distribution and removal from the atmosphere. This interdependency of aerosol and clouds mediated by local meteorology plays a crucial role in climate regulation, radiative forcing and the distribution of precipitation. However, even after decades of research, a full grasp of the nature of aerosol–cloud interactions remains one of the big challenges and constitutes one of the largest sources of uncertainty in our understanding of climate forcing and feedbacks (Solomon et al., 2007; Quaas et al., 2009; Carslaw et al., 2010; Bellouin et al., 2019).

The vast range of spatio-temporal scales at which aerosol–cloud interactions occur helps us to appreciate why it is so difficult to pin down their role in the climate system. For example, processes such as condensation and sublimation occur at nano- to micrometre spatial scales, while the interplay between aerosols and clouds over the major pollution outflow regions occurs at much larger spatial scales. The temporal component of aerosol–cloud interactions also stretches from microseconds, to days and weeks of persistent pollution outflow, to decades of emission policy changes. To account for these wide ranges of spatio-temporal scales, harmonization and representation in a physically consistent and observationally constrained manner in global climate models are extremely challenging. At the same time, the evaluation of climate models must cover these different spatio-temporal scales. This can help us to assess, among others, at what spatio-temporal scales the aerosol–cloud interactions can distinctly manifest themselves and could even dominate the local variability.

In this context, this study focuses on understanding variability at larger scales. We specifically aim to understand the co-variability between AOD and CD effective radius at the climate scale. We acknowledge that both AOD and CD effective radius can be independently influenced by a number of processes and that CCN number concentration, especially at the cloud base, is better suited to investigate aerosol impact on cloud albedo via changes in droplet radius, as discussed

in a number of previous studies (e.g. Quaas et al., 2020, and the references therein). The correspondence between AOD and CCN number concentration has been observed in a number of studies. Andreae (2009) identified a correlation between AOD and CCN concentration by comparing AERONET AOD with CCN measurements. Romakkaniemi et al. (2012) suggested the potential use of AOD as a proxy for CCN concentration. While uncertainties persist, Tang et al. (2014) noted the limitations of AOD as an indicator of CCN, acknowledging variations in CCN concentrations for a given AOD based on regional characteristics and meteorological conditions, such as relative humidity. Despite these complexities, Andreae (2009) observed a high correlation coefficient of 0.98 between AOD and CCN concentration. Using the ground measurements of CCN at the Zeppelin Observatory (78.91° N, 11.89° E), Ahn et al. (2021) derived a CCN–AOD correlation with a coefficient of determination $R^2$ of 0.59.

We choose AOD and CD effective radius here because these variables are better grounded in observations and can be evaluated more directly using satellite-based observations. Furthermore, we assume that if aerosols indeed are one of the primary drivers of changes in cloud properties, especially under favourable meteorological conditions (Fanourgakis et al., 2019), their co-variability at much larger temporal and spatial scales will visibly manifest itself in a relation between AOD and CD effective radius.

In this study, we first evaluate the recent improvements made in the EC-Earth3-AerChem model using a suite of satellite-based observations, and we then employ maximum co-variance analysis on AOD and CD effective radius to explore the intricate interplay between aerosols and clouds. We primarily focus on the oceanic regions that sustain low-level liquid water clouds and where the pollution outflow also occurs at least to some degree. The net cooling effect of low-level liquid water clouds over open oceans is very important, and their susceptibility to aerosols has significant implications for the energy budget.

Within the framework of the FORCES project (https://forces-project.eu/, last access: 5 March 2024), this study aims to identify and implement the crucial aerosol and cloud processes that could potentially reduce uncertainties in aerosol forcing estimates. This is detailed in Sect. 2.2. These improvements are being incorporated into the three global models that were previously used for CMIP6 (https://pcmdi.llnl.gov/CMIP6/, last access: 13 February 2019): NorESM2-LM (Seland et al., 2020), MPI-ESM1.2-HAM (Mauritsen et al., 2019; Neubauer et al., 2019) and EC-Earth3-AerChem (van Noije et al., 2021). This paper focuses on the simulations carried out with the EC-Earth3-AerChem model.

## 2 Model, observations and methods

### 2.1 EC-Earth3-AerChem model description

The EC-Earth3-AerChem model is based on the EC-Earth3 family of models (Döscher et al., 2022). The atmospheric component of the model is based on IFS cycle 36r4, which includes the land surface model H-TESSEL (Balsamo et al., 2009). The resolution of the atmospheric model is T255L91 with a grid spacing of approximately 80 km and 91 levels in the vertical level that extends to 0.01 hPa. The AerChem configuration features the aerosol and chemistry model Tracer Model version 5 (TM5) (Krol et al., 2005; Huijnen et al., 2010). This chemistry and aerosol module is computationally expensive and is run at a much coarser resolution of $3° \times 2°$ with 34 vertical levels compared with the atmospheric model. The cloud macrophysics and microphysics are part of the atmosphere model and so are run at the same T255L91 resolution. The details of the CMIP6 version of the EC-Earth3-AerChem model are described in van Noije et al. (2021).

### 2.2 EC-Earth3-AerChem model updates within FORCeS

The FORCeS project aimed at the improvement of various crucial processes in the global climate models that would have the potential to influence aerosol radiative forcing and climate feedbacks, particularly concerning aerosols and clouds so as to improve our future climate projections. In this context, the different minerals present in dust aerosols are now explicitly traced, and their interaction with specific climate processes has been implemented and the cloud activation scheme was updated. The following paragraphs give a more detailed but brief overview of these updates.

### 2.2.1 Cloud activation scheme

The formation of cloud droplets from ambient aerosol particles is a crucial process that needs to be accurately represented in global models to predict aerosol–cloud interactions and, hence, the indirect effects of aerosols on climate. To achieve this, the modelling community has developed parameterization schemes that can realistically simulate the cloud droplet number concentrations based on ambient supersaturation, aerosol size and number distribution, and updraft velocities. The schemes range from simpler schemes such as Abdul-Razzak and Ghan (2000) and Shipway (2015) to more complex schemes such as Barahona et al. (2010) and Morales-Betancourt and Nenes (2014). All parameterizations show good overall agreement with the parcel model but exhibit different sensitivities to aerosol perturbations under different meteorological conditions or pollution regimes (Ghan et al., 2011). For example, the Abdul-Razzak and Ghan (2000) scheme simulates low/high maximum supersaturations ($S_{max}$) and activation at low/high aerosol loads compared with the Morales-Betancourt and Nenes (2014)

scheme, making it more sensitive to increases in aerosol loads. The Shipway (2015) scheme is closer to the Morales-Betancourt and Nenes (2014) scheme but tends to overestimate $S_{max}$ and activation in polluted environments. Also, the simpler schemes show larger biases than the complex schemes in clean marine regions, particularly over the Southern Ocean. Hence, the current cloud activation scheme of Abdul-Razzak and Ghan (2000) is replaced by the Morales-Betancourt and Nenes (2014) scheme in the FORCeS version of the EC-Earth3-AerChem model. This scheme makes use of the population splitting concept wherein the growing population of droplets is divided into 3, enabling a more accurate estimate for the number concentration of droplets formed at the time of $S_{max}$. TS1

### 2.2.2 Dust mineralogy

EC-Earth3-AerChem calculates dust emission online (van Noije et al., 2021). The emission scheme considers dry paleolakes as preferential dust sources and accounts for seasonal variations in vegetation cover. Saltation of dust particles occurs when the surface wind exceeds a certain threshold velocity, which depends on the soil particle size distribution and soil conditions. Furthermore, snow cover prevents dust emission in the model (Tegen et al., 2002, 2004). Dust is initially emitted into the accumulation and coarse insoluble modes, defined in the M7 aerosol microphysics module (Vignati et al., 2004). Within FORCeS, the dust emission module has been further developed to explicitly incorporate the atmospheric cycle of dust minerals relevant for their climate impacts. These include (1) iron oxides, which control the absorption of shortwave (SW) radiation by dust aerosols (e.g. Sokolik and Toon, 1999; Di Biagio et al., 2019); (2) quartz and feldspars, which constitute efficient ice nucleators (e.g. Atkinson et al., 2013; Harrison et al., 2019); and (3) calcite, which intervenes in atmospheric chemistry processes and affects the aerosol pH.

To estimate the emission of the different minerals, we derive the size-distributed mass fraction at emission for each of them considering information on the soil mineralogy of the dust sources (Claquin et al., 1999; Nickovic et al., 2012). To account for the differences in the mineral size distribution reported in the soil and those found in the aerosols, we apply an extension of brittle fragmentation theory for dust emission proposed by Kok (2011). This approach has shown a better agreement with observations (e.g. Perlwitz et al., 2015a, b) than relying exclusively on the soil size fractions. These newly incorporated minerals are then coupled with different processes in the model. The effect of iron oxides in the SW optical properties for dust is accounted for by assuming internal mixtures of iron oxides and other host minerals. A Maxwell Garnett mixing rule is applied to derive online the dust refractive index, considering the volume fraction of hematite and a homogeneous matrix of other minerals. The reference refractive indices are taken from Scanza

et al. (2015). The abundance of quartz and K-feldspars is considered to derive the ice nucleating particles and then coupled with the formation of mixed-phase clouds in the model. In order to estimate the specific fraction of K-feldspar, we assume it to be 35 % of the total feldspar originally provided in the soil mineralogy datasets (Atkinson et al., 2013; Chatziparaschos et al., 2023). Finally, the calcite in dust is explicitly linked to the thermodynamic equilibrium of inorganic species, solved in the model with the ISORROPIA-Lite (Kakavas et al., 2022) model (see Sect. 2.2.5).

### 2.2.3 MPOA source

Marine organic aerosols have been suggested as a relevant source of ice-nucleating particles, particularly in remote marine environments (Wilson et al., 2015). The EC-Earth3-AerChem FORCeS model version includes a new source of marine primary organic aerosols (MPOAs), which considers the partitioning between insoluble marine organics and sea salt. The MPOA emission is calculated as a fraction of the submicron sea salt aerosols dependent on the chlorophyll $a$ (Chl $a$) present in the ocean surface layer (O'Dowd et al., 2008; Vignati et al., 2010). In addition, a coarse-mode MPOA is included (Facchini et al., 2008; Myriokefalitakis et al., 2010), which also depends on the online calculated sea salt emission (Gong, 2003). The model uses Chl $a$ concentration, a monthly averaged product derived from MODIS (Moderate Resolution Imaging Spectroradiometer) satellite observations.

### 2.2.4 Primary and secondary ice production

In regions such as the remote Arctic/Antarctica where the primary ice nucleating particles (INPs) are sparse (Wex et al., 2019; Moore et al., 2024), a considerable source of ice crystals is via secondary ice production (SIP) processes in supercooled clouds (temperatures higher than approximately $-10\,°C$) (Järvinen et al., 2022, 2023). This process by which ice crystals are generated outnumbers those generated by primary INPs (Field et al., 2017). The mechanisms that contribute to the INP formation are not adequately represented in the models, resulting in a significant underestimation of the observed ice crystal number concentrations, which in turn would impact the radiative forcing estimates (Vergara-Temprado et al., 2018; Zhao et al., 2023). In this project, an aerosol- and temperature-sensitive parameterization (Costa-Surós et al., 2023) has been developed to substitute the temperature-based parameterization by Meyers et al. (1992) for estimating the ice crystal number concentrations (ICNCs) in mixed-phase clouds. Specifically, it considers the primary ice crystal formation by immersion freezing dependent on K-feldspar and quartz (Harrison et al., 2019), deposition nucleation of soot and dust (Ullrich et al., 2017), and immersion freezing of marine organic aerosols (Wilson et al., 2015). The ICNCs formed via SIP are quantified using the RaF-SIP (Random Forest Secondary Ice Production) scheme version 1, as detailed in Georgakaki and Nenes (2024). RaFSIP is a data-driven parameterization developed from a 2-year simulation with a 10 km horizontal grid spacing, covering the period from 2016 to 2017 over the pan-Arctic region using the Weather Research and Forecasting (WRF) model (https://www.mmm.ucar.edu/models/wrf, last access: 22 August 2024) with explicit SIP microphysics (Sotiropoulou et al., 2021; Georgakaki et al., 2022). Unlike conventional SIP parameterizations (e.g. Phillips et al., 2017), the RaFSIP scheme requires only a few input variables, offering a streamlined and easily implementable solution for large-scale models without detailed microphysics schemes, which would not otherwise support the explicit treatment of SIP.

The effect of SIP through collisional break-up, droplet shattering and Hallett–Mossop rime splintering is described in the model through the introduction of an ice enhancement factor (IEF) – a multiplication factor applied to primary ice production rates. The notion of the IEF is often used in the literature to indicate the prevalence of SIP. This approach shows promise for parameterizing SIP effects in mixed-phase clouds, as it enables direct comparison with IEF (ICNCs/INPs) derived from in situ observations or retrieved from remote sensing data (e.g. Wieder et al., 2022).

### 2.2.5 ISORROPIA-Lite for inorganic aerosols

A new thermodynamic module for inorganic aerosols, ISORROPIA-Lite (Kakavas et al., 2022), is also implemented in the updated EC-Earth3-AerChem model replacing EQSAM (Metzger et al., 2002). ISORROPIA-lite is based on the ISORROPIA-II (Fountoukis and Nenes, 2007) code, and it treats the thermodynamics of aerosol containing $Ca^{2+}$, $K^+$, $Mg^{2+}$, $SO_4^{2-}$, $Na^+$, $NH_4^+$, $NO_3^-$, $Cl^-$ and $H_2O$ and their equilibrium with gas-phase $HNO_3$, $NH_3$, $HCl$ and $H_2O$. Furthermore, the bulk calculation of ammonium nitrate has been replaced by one which distributes these components over the accumulation and coarse soluble modes.

### 2.2.6 Model tuning

The model undergoes a tuning process after major updates to minimize the radiative imbalance at the top of the atmosphere (TOA) and at the surface (global mean net radiative flux at the TOA, TOA and surface SW and LW flux, cloud radiative effects in LW and SW) in AMIP simulations. This involves adjusting specific parameters with respect to the CERES-EBAF dataset (see Sect. 2.3) that is used here as observational reference. In this study, the following three key parameters are adjusted for the tuning of the EC-Earth3-AerChem model:

- The conversion efficiency parameter, RPRCON, which determines the rate of autoconversion of cloud water to rain, is increased from $1.34 \times 10^{-3}$ to $1.41 \times 10^{-3}$.

- The critical radius for autoconversion is reduced from 8.75 to 7.5 μm.

- The standard deviation of the updraft velocity distribution is reduced by $0.2\,\mathrm{m\,s^{-1}}$ (from 0.8 to $0.6\,\mathrm{m\,s^{-1}}$) in the cloud droplet activation process.

Prior to these adjustments, the combined model updates resulted in a TOA net radiation imbalance of more than $-3\,\mathrm{W\,m^{-2}}$ in comparison with the CMIP6 version of the model, which had an imbalance of $-0.46\,\mathrm{W\,m^{-2}}$. By fine-tuning the aforementioned parameters, the model is closer to a radiative balance with a net TOA imbalance of $-0.7\,\mathrm{W\,m^{-2}}$.

In the following, we will refer to the CMIP6 configuration of the EC-Earth3-AerChem model as "ECE3-CMIP6", while the configuration including FORCeS updates will be referred to as "ECE3-FORCeS".

## 2.3 Observational datasets

To evaluate the impacts of the model updates detailed in Sect. 3, we use a suite of satellite-based datasets of cloud and aerosol properties. These are described below.

*MODIS-Aqua*. Retrievals of cloud properties from the MODIS on board the Aqua satellite in the framework of NASA's Earth Observing System (EOS) are used. We specifically use Collection 6 Level-2 and Level-3 (MYD06_L2 and MYD08_L3) information on cloud fraction, optical thickness, liquid water path and droplet effective radius. We use 18 years of data from 2003 to 2020.

*CERES-EBAF*. The radiative flux components from the Clouds and the Earth's Radiant Energy System (CERES) Energy Balanced and Filled (EBAF) dataset are used to evaluate the top of the atmosphere shortwave and longwave fluxes. The latest Edition 4.2 data at Level-3b are used for the analysis (Loeb et al., 2018; Kato et al., 2018). The period 2000–2016 is made use of in this study.

*CPR-CloudSat*. The latest Release 5 of CloudSat Level 2B-CWC-RVOD and 2B-CWC-RO (CloudSat Radar-Only Cloud Water Content) products are used to evaluate the cloud liquid water path (CLWP) and cloud ice water path (CIWP) respectively. These products are derived for each radar profile as seen by CloudSat's Cloud Profiling Radar (CPR) for those profiles for which clouds are likely based on the radar profile analysis (https://www.cloudsat.cira.colostate.edu/cloudsat-static/info/dl/2b-cwc-ro/2B-CWC-RO_PDICD.P1_R05.rev0_.pdf, last access: 30 January 2023). In the 2B-CWC-RVOD product, the retrievals of CLWP are constrained using cloud optical depth information from MODIS (https://www.cloudsat.cira.colostate.edu/data-products/2b-cwc-rvod, last access: 3 March 2019). Unlike the passive sensors, the CPR on board CloudSat can sense the entire cloud column, day and night, thus providing better estimates of CLWP and CIWP. We use the data from 2007 to 2011.

*CALIPSO*. We further use the Cloud-Aerosol Lidar and Infrared Pathfinder Satellite Observations (CALIPSO) Lidar Level-3 Global Energy and Water Cycle Experiment (GEWEX) Cloud, Standard Version 1-00 data product (CAL_LID_L3_GEWEX_Cl oud-Standard-V1-00) to obtain information on total cloud fraction and its subdivision into low, medium and high cloud fraction (NASA/LARC/SD/ASDC, 2019). This latest Level-3 version is based on the version 4.20 Level-2 5 km merged layer product for the period 2007–2016.

The climatologies of cloud and aerosol properties from these datasets averaged over their respective time periods are statistically compared with the model simulations in Sect. 3.

## 2.4 Maximum covariance analysis: description and methodology

Maximum covariance analysis (MCA) (Bretherton et al., 1992; Cherry, 1996) is a statistical technique that extracts coherent patterns in two datasets that explain the maximum fraction of the covariance between them. This means that the analysis identifies the regions where the two fields co-vary to a maximum extent in the different spatial modes. The resulting modes are orthogonal to each other, making the patterns complex and difficult to interpret physically. Hence, "rotated" MCA is employed here to maximize the separation between the patterns by relaxing the orthogonality constraint, thereby improving their interpretability. A varimax rotation technique is used in this study. This analysis provides two maps: (1) a homogeneous regression map and (2) a heterogeneous regression map. The homogeneous patterns are the correlation coefficients between the input data of one field and scores (see the next section on how scores are calculated) of the same field, whereas the heterogeneous patterns are the correlation coefficients between the input data of one field and scores of the second field and vice versa. The resulting heterogeneous maps are a typical characteristic of the MCA, as they bring out the covariability between the two datasets. The scores are similar to the principal components (PCs) that capture the amplitude and temporal variation or, in other words, the variability associated with each spatial pattern (or mode). The squared covariance fraction (SCF) is an invariant quantity even if the modes are modified or transformed in some way. This fraction assesses the relative importance of each mode.

### 2.4.1 Mathematical background

A brief overview of the MC analysis is given here.

Assume two datasets, $X$ and $Y$, with variables $a$ and $b$ respectively, which can be represented as matrices, $X(a \times n)$ and $Y(b \times n)$, where $n$ is the number of observations. Both $X$ and $Y$ are first standardized and detrended.

The scores are defined as projections of $X$ and $Y$ on to the singular vectors of the cross-covariance matrix of $X$ and

$Y$. The cross-covariance matrix $C_{xy}$ is computed as in the equation below.

$$C_{xy} = \left(\frac{1}{n}\right) XY^T,$$

where $Y^T$ is the transpose matrix of $Y$.

A singular value decomposition is performed on the cross-covariance matrix as

$$C_{xy} = \mathbf{U}\mathbf{D}\mathbf{V}^T,$$

where $U$ and $V$ are correspondingly the eigenvectors for the two datasets, $X$ and $Y$, and $\mathbf{D}$ is a diagonal matrix with singular values.

The scores are then calculated as

$$A_x = U^T X,$$
$$A_y = V^T Y.$$

The homogeneous patterns for $X$ and $Y$ fields are defined as

$$\text{hom}_X = \text{Corr}(X, A_x),$$
$$\text{hom}_Y = \text{Corr}(Y, A_y).$$

Similarly, the heterogeneous patterns for $X$ and $Y$ fields are defined as

$$\text{het}_X = \text{Corr}(X, A_y),$$
$$\text{het}_Y = \text{Corr}(Y, A_x),$$

where "Corr" refers to the correlation coefficient between the two fields within the brackets which is calculated as in the equation below.

For example,

$$\text{Corr}(X, A_x) = \frac{\sum (X - \bar{X})(A_x - \bar{A}_x)}{\sqrt{\sum (X - \bar{X})^2 \sum (A_x - \bar{A}_x)^2}},$$

where the summation $\sum$ is the summation over the total number of observations.

The squared covariance fraction quantifies the extent to which each mode $i$ accounts for the explained proportion of the total squared covariance and is defined as

$$\text{SCF}_i = \frac{\sigma_i^2}{\sum_{i=1}^m \sigma_i^2},$$

where $m$ is the total number of modes and $\sigma_i$ is the $i$th singular value of the covariance matrix.

## 3 Results I: evaluation of the recent improvements in ECE3-FORCeS model

Historical atmosphere-only (AMIP) FORCeS simulations for the period 1980–2020 are used for the analysis. The comparisons are made with the corresponding CMIP6 simulations that cover a period from 1980 to 2018. Both AMIP simulations are fully compliant with the CMIP6 AMIP protocol but have been extended beyond the CMIP6 historical period using anthropogenic and biomass burning emissions of ozone and aerosol precursors, greenhouse gas concentrations and other forcings following the CMIP6 protocol for the SSP2-4.5 scenario experiment. Sea surface temperatures (SSTs) and sea ice fractions are prescribed using the CMIP6 version (1.1.8) of the PCMDI forcing dataset (https://www.wdc-climate.de/ui/cmip6?input=input4MIPs. CMIP6.CMIP.PCMDI.PCMDI-AMIP-1-1-8.ocean.mon. tosbcs.gn.v20220622, last access: 22 June 2022), which extends through 2021. The forcing dataset is based on the UK MetOffice HadISST and NCEP OI2 (Durack et al., 2022).

To facilitate the evaluation of the model-simulated variables against satellite observations in a consistent manner, the COSP (Cloud Feedback Model Intercomparison Project (CFMIP) Observation Simulator Package) (Bodas-Salcedo et al., 2011) that is developed by the CFMIP community is available in EC-Earth3-AerChem. COSP provides tools and algorithms to simulate what a climate model would "see" from the viewpoint of the satellite that is used. This allows for a more direct comparison with satellite observations (Pincus et al., 2012).

In the following sections, we focus on the evaluation of the ECE3-FORCeS version and provide a comparison against both the ECE-CMIP6 version and the observations.

### 3.1 Evaluation of COSP-simulated cloud fractions

In this section, simulated cloud and aerosol from the ECE3-FORCeS version of the model are evaluated against satellite observations and compared against the simulations of ECE3-CMIP6 version of the model. Here, the simulated cloud parameters derived from the COSP simulator of the model are evaluated.

The EC-Earth3-AerChem model outputs the following COSP parameters: CALIPSO-COSP total, low, middle and high cloud fractions and MODIS-COSP total, water and ice cloud fractions. The CALIPSO-COSP-simulated climatological mean total and the cloud fractions at three altitudes are presented in Fig. 1b. To facilitate the evaluation, CALIPSO observations (a) and the differences of the simulated values with respect to CALIPSO observations (c) and CMIP6 simulations (d) are also shown. The values in brackets correspond to the global mean. The ECE3-FORCeS model simulates the spatial distribution of the total cloud fraction reasonably well compared with the observations, with an overestimation in the polar latitudes. This overestimation stems from the biases which are primarily seen in the low cloud fraction. The high cloud fraction, on the other hand, is considerably underestimated globally with respect to the observations particularly over the oceanic regions west of South America, Africa

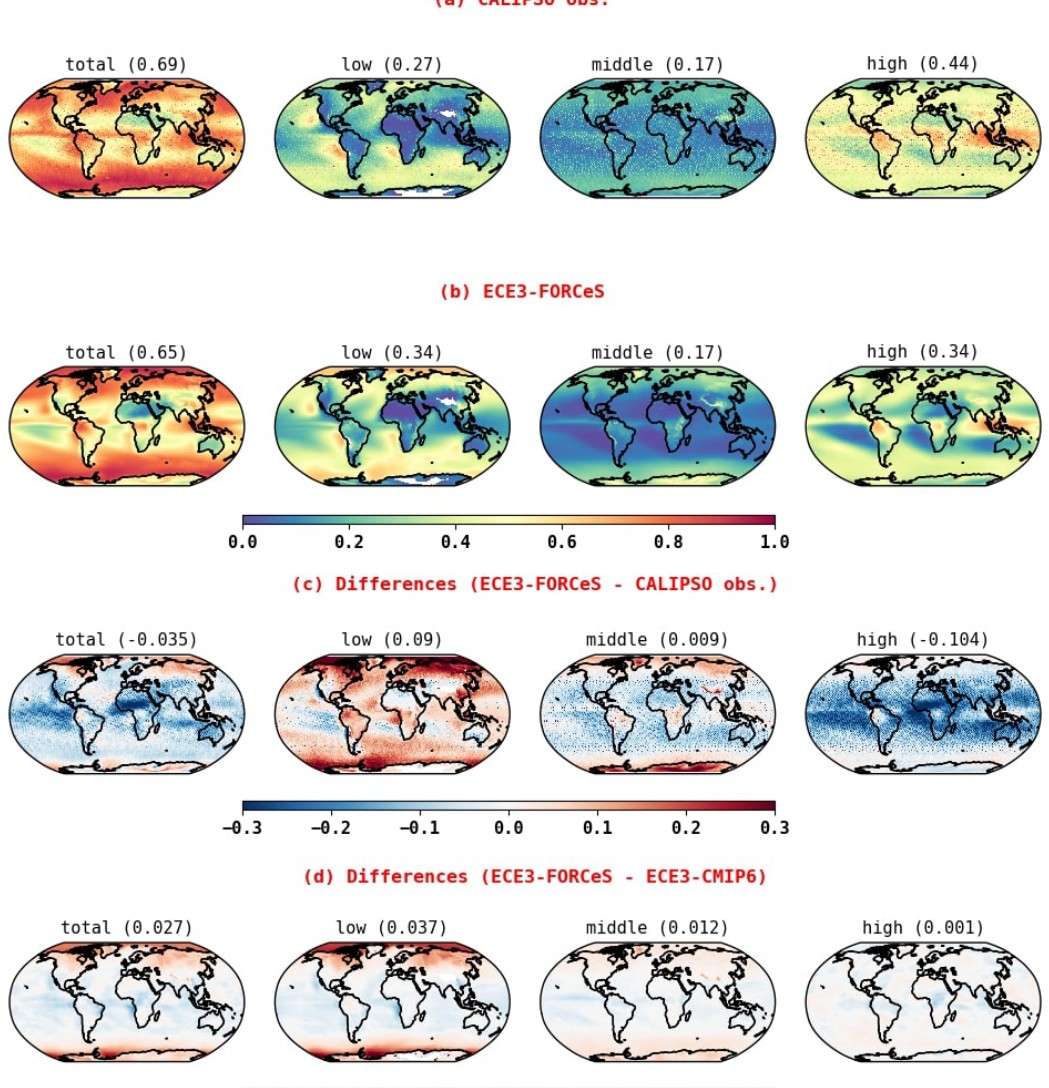

**Figure 1.** Climatological mean total, low, middle and high cloud fractions respectively in the four columns from **(a)** CALIPSO observations and **(b)** COSP-CALIPSO simulations from ECE3-FORCeS and their differences with respect to **(c)** CALIPSO observations and **(d)** ECE3-CMIP6 simulations. The values in the brackets refer to the global mean of each category. The observations extend from 83° S to 83° N.

and Australia. This explains the underestimation of the total cloud fraction over the oceanic regions by 20 %–30 % with respect to the observations. The simulated global mean total cloud fraction is 65 %, whereas the observed mean fraction is 69 %. The aerosol and cloud processes have resulted in an increase in the total cloud fraction in the high latitudes in both hemispheres with respect to the corresponding ECE3-CMIP6 simulations, and this increase can be attributed to the increase in low cloud fraction and partly to an increase in middle cloud fraction. This increase in low cloud fraction can be attributed to updates in the model, such as the cloud droplet activation that targets liquid water clouds. No signif-

icant change is seen in the tropics and mid-latitudes with the model updates.

The ECE3-FORCeS and ECE3-CMIP6 zonal averaged climatological mean total, low, middle and high cloud fractions are evaluated against CALIPSO observations (see Fig. 2a–d). The solid black lines in each category of the cloud fraction refer to the observed value. The total cloud fraction in the Southern Hemisphere (SH) high latitudes is much closer to the observations in the ECE3-FORCeS model. It can be clearly seen that the improvements in the ECE3-FORCeS version of the model do not bring a significant change in the cloud fraction over the tropics and mid-

latitudes compared with its ECE3-CMIP6 counterpart. However, the total and the low cloud fractions in ECE3-FORCeS simulations in the Northern Hemisphere (NH) high latitudes tend to deviate even further from the observations compared with those in ECE3-CMIP6 runs.

A comparison of the simulated climatological mean total, water and ice cloud fractions with MODIS observations (Fig. 3) shows that although the model simulates the spatial distribution of total cloud fraction well, it fails to reproduce the magnitude globally, particularly over the tropics and mid-latitudes, as can be clearly seen in the difference plots (Fig. 3c). The model captures the liquid water cloud fraction more realistically, and, over the aforementioned regions, there is a slight overestimation of about 10 %–15 % compared with the observations. The global mean values of the water cloud fraction are close to 30 % in both the observations and simulations. The ice cloud fraction, on the other hand, is overestimated over the mid-latitudes and high latitudes in both hemispheres in the model, thereby overestimating the global mean values of the ice cloud fraction by 8 %. It can be noted that the simulated ice cloud fraction over the stratocumulus cloud decks is comparable to what is seen by MODIS. This would indicate that the mixed cloud fraction in the model is underestimated, which could partly explain the underestimation seen in the simulated total cloud fraction across these regions.

## 3.2  Evaluation of cloud radiative effects

The cloud radiative effects (CREs) at the TOA are defined as the difference between the TOA all-sky and clear-sky fluxes. The spatial distribution and zonal means of the simulated climatological mean shortwave (SW) and longwave (LW) TOA CRE are evaluated against CERES-EBAF observations, and the results are illustrated in Fig. 4 as differences in the simulated values from the observed values. The global means are shown in brackets. The overestimation of the SW cloud forcing in the ECE3-FORCeS simulations over the Southern Ocean and along the Equator and over the pollutant outflow regions of Africa, North America, South America and eastern China is more than halved compared with the ECE3-CMIP6 simulations where the biases were more than $40 \, \text{W m}^{-2}$ (Fig. 4, column 1). The global mean bias in the SW CRE is only one-tenth in the ECE3-FORCeS model compared with its previous model version.

On the other hand, the LW CREs are underestimated, particularly along the equatorial belt (Fig. 4, column 2). The slight positive bias in LW in ECE3-CMIP6 simulations in the stratocumulus regions over the western coast of the continents is reduced in ECE3-FORCeS. However, we overestimate the LW forcing in the NH high latitudes and over the Southern Ocean in the ECE3-FORCeS model. The global mean LW CRE shifted from a negative bias ($-0.96$) to a slightly more positive bias ($0.55$). There is a considerable improvement in the SW CRE with the FORCeS model updates,

which can be seen clearly in the zonal mean plots shown to the right in Fig. 4. The overestimation in the SW CRE at the TOA is notably reduced by more than $20 \, \text{W m}^{-2}$ over the Southern Ocean. By contrast, there is an overestimation in the LW CRE at latitudes of $> 50°$ in the FORCeS version, with biases in the range of $3–5 \, \text{W m}^{-2}$, while in the mid-latitudes (between $50°$ S and $50°$ N), there is a slight improvement.

## 3.3  Evaluation of cloud microphysical properties

Measuring cloud water is crucial for investigating microphysical processes and the indirect impacts of aerosols. Cloud water is commonly represented using column-integrated metrics such as cloud liquid water path and cloud ice water path. Since the COSP simulator for CloudSat products is not available, we use the standard model output for CLWP and CIWP for comparison with CloudSat retrievals. This comparison could still be reasonable, as CloudSat can see through the whole cloudy column. Furthermore, using the MODIS visible optical depth constrained product for CLWP and Radar-Only product for CIWP provides the best reference for cloud water free from the contamination by precipitation signal.

The spatial distribution of the ECE3-FORCeS-simulated climatological mean fields against CloudSat observations and the differences from ECE3-CMIP6 simulations are presented in Fig. 5. These maps need to be interpreted with caution, as the colour bar range varies between the model and the observations. Despite the model's realistic simulation of spatial distribution, both CLWP and CIWP are markedly underestimated when compared with the actual observations. Following the FORCeS updates, CLWP increased over high latitudes and mid-latitudes in both hemispheres, while it decreased over the tropics compared with the ECE3-CMIP6 version of the model. A significant global decrease of up to $3.5 \, \text{g m}^{-2}$ in CIWP is simulated in the ECE3-FORCeS model with regional decreases of up to $10–15 \, \text{g m}^{-2}$, particularly over the oceans in the NH and SH.

The simulated climatological mean CD effective radius is evaluated against MODIS observations as in Fig. 6. The global mean values are shown in the brackets for the same region ($75°$ S–$75°$ N) as for the MODIS observations. The magnitude and spatial pattern along the equatorial oceans are very close to the observations. However, the droplet sizes are comparatively smaller in the simulations than in the observations. The observed global climatological mean is $14.7 \, \mu\text{m}$ compared with a value of $7 \, \mu\text{m}$ in the ECE3-FORCeS model simulation. The differences in the CD effective radius are in the range of uncertainty of the observations, wherein the biases in regional monthly mean MODIS-derived values are at least $1–10 \, \mu\text{m}$ depending on cloud horizontal heterogeneity and solar zenith angle (Fu et al., 2019). Notably, the latest model updates have led to an increase in CD effective radius,

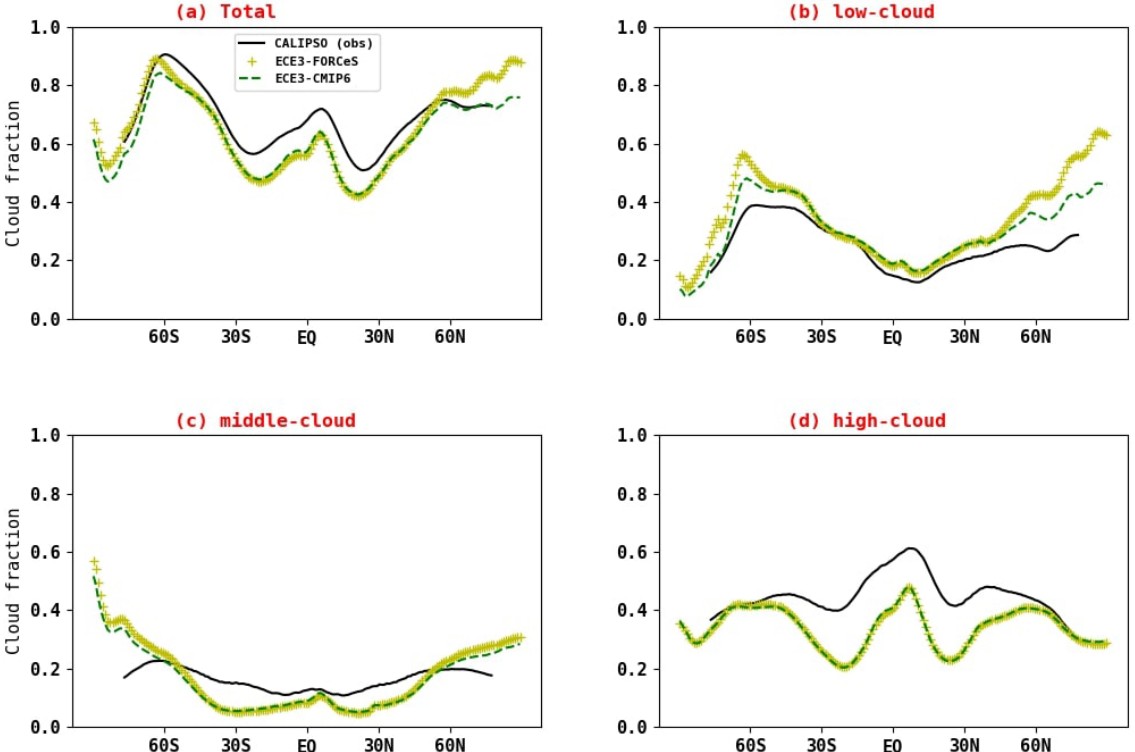

**Figure 2.** Zonally averaged climatological mean CALIPSO-COSP simulated **(a)** total, **(b)** low, **(c)** middle and **(d)** high cloud fractions. Solid black lines correspond to the CALIPSO observations. The dashed green lines are the ECE3-CMIP6 simulations, and the dotted yellow lines correspond to the ECE3-FORCeS simulations.

especially over the tropical oceans, as observed in comparison with the ECE3-CMIP6 simulations.

## 4 Results II: maximum covariance analysis – covariability between AOD and CD effective radius

### 4.1 Regions selected for the analysis

In order to investigate the maximum covariance between AOD and CD effective radius, we select the regions based on two prerequisites: (a) the regions sustain low-level liquid water clouds, and (b) the regions experience at least some degree of pollution outflow from the continents. Among all the oceanic cloud regimes, these low-level clouds are most likely to be influenced by aerosols and stand the highest chance of testing the hypothesis that aerosol–cloud covariability can also be visible in the larger spatio-temporal scales. We selected three such regions based on our previous study (Devasthale and Thomas, 2011), which investigated the frequency of occurrence of aerosol–cloud overlap globally in different seasons. They are shown in Fig. 7 by the blue boxes.

Table 1 provides details about the selected regions, each distinguished by specific types of aerosols. For instance, the

**Table 1.** Regions selected for the study. "BB" refers to biomass burning.

| Regions | Latitude, longitude | Time periods |
|---|---|---|
| Eastern China | 20–50° N, 95–150° E | All months |
| BB1: Africa | 25–5° S, 10° W–20° E | Sep–Nov |
| BB2: South America | 15° S–5° N, 100–70° W | Jun–Nov |

eastern China region is marked by anthropogenic aerosols, persisting throughout the year but with a reduced intensity during the June–July–August season due to pollutant removal by wet deposition. The remaining two regions, designated as BB1 and BB2 respectively, for the west coast of Africa and the west coast of South America, are notably influenced by biomass burning aerosols in the free troposphere and marine aerosols in the boundary layer (Bourgeois et al., 2015, 2018). The prevalence of transported biomass burning aerosols is higher during September–November for Africa and June–November for South America during the dry season.

The climatological mean AOD and low cloud fraction for the three regions selected for this study are shown in Fig. 8.

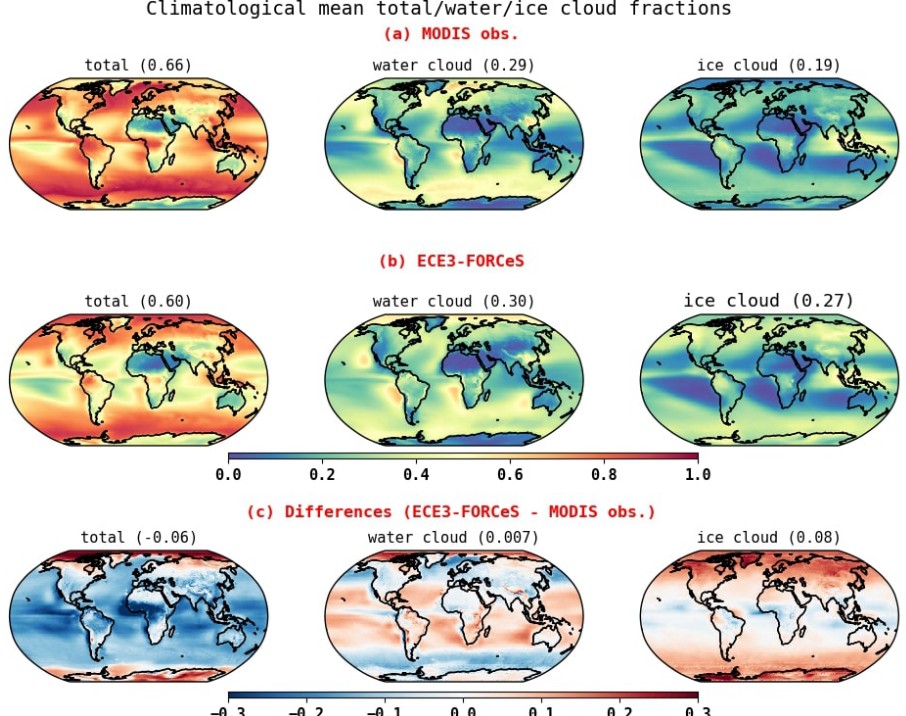

**Figure 3.** Climatological mean total (left), liquid water (middle) and ice (right) cloud fractions respectively in the three columns from **(a)** MODIS observations and **(b)** ECE3-FORCeS simulations and **(c)** their difference with respect to MODIS observations.

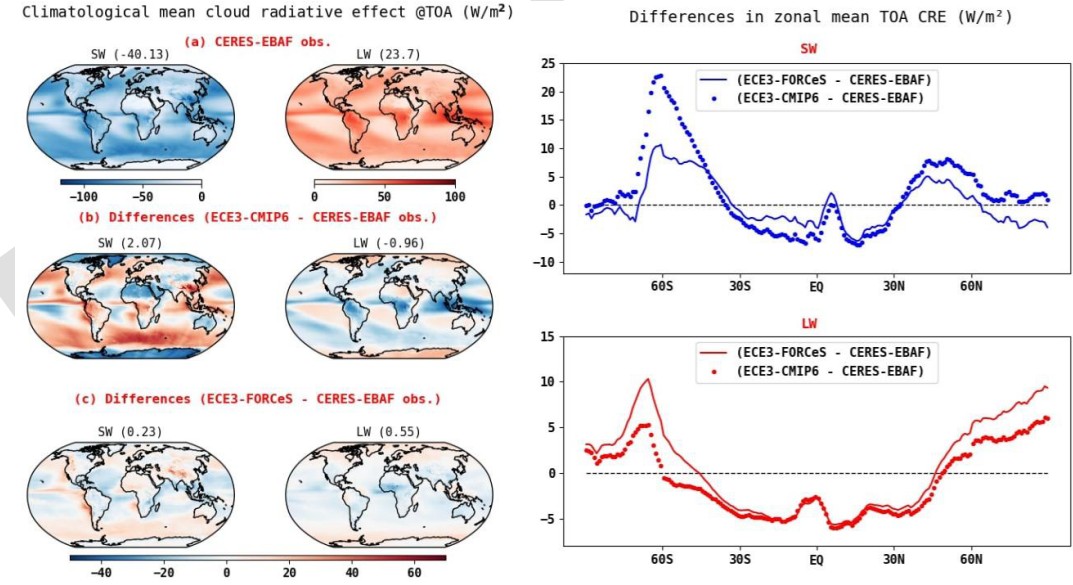

**Figure 4.** Left: climatological mean CRE (in $W\,m^{-2}$) in SW (left) and LW (right) at the TOA from **(a)** CERES-EBAF observations and **(b)** differences in CMIP6 simulations and **(c)** ECE3-FORCeS simulations from observations. Right: zonally averaged climatological mean CRE in SW and LW shown as differences from the observations.

Among these regions, eastern China experiences the highest AOD, reaching 0.7, in the area of maximum anthropogenic pollution. In the other two regions, AOD peaks over land with values around 0.25, corresponding to areas where biomass burning occurs. The Andes mountain range, situated in the west of the South American continent, acts as a barrier that partially hinders the transport of pollutants to the oceans. The low cloud fraction ranges from 0.3 to 0.7 over these

Climatological mean cloud liquid and ice paths (g/m²)

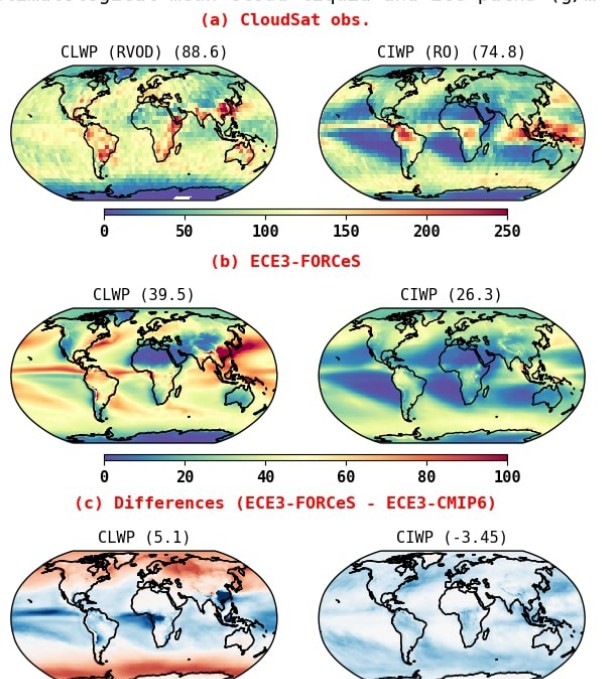

**Figure 5.** Climatological mean cloud liquid water (left) and ice (right) paths $(\mathrm{g\,m^{-2}})$ from **(a)** CloudSat observations and **(b)** ECE3-FORCeS simulations and **(c)** their differences with respect to ECE3-CMIP6 simulations. The global mean values are given in brackets.

regions. It should also be noted that the regions where the maximum pollution outflow over the oceans occurs and the regions where the maximum frequency of low-level liquid clouds is observed are not the same. The maximum aerosol–cloud overlap in fact occurs over the regions more remote from the individual maxima in AOD and cloud fraction.

### 4.2 MC analysis over eastern China

As explained in Sect. 2.4, maximum covariance analysis is applied to assess the covariability between AOD and CD effective radius across eastern China. Before applying the MCA, both these variables are de-seasonalized and detrended. The heterogeneous patterns, the patterns that bring out the areas of covariability between the two fields, are presented. Only regions exhibiting statistical significance at the 95 % confidence level are showcased. We present the first three modes derived from the analysis for eastern China in Fig. 9a–c. These three modes collectively explain 66 % of the total observed variability. A distinct pattern exhibiting opposite signs between AOD (left) and CD effective radius (right) emerges, and each mode captures the different regions where this inverse covariability is dominant. The opposite signature between these two fields clearly suggests that the Twomey

effect over those regions in the corresponding modes can influence the variability at the large spatio-temporal scales. The first two modes capture the covariability over the outflow regions where the aerosol advection over nearby oceanic low-level clouds is quite frequent (Devasthale and Thomas, 2011), whereas the third mode captures the covariability over the continental source regions. It is intriguing that the covariability between AOD and CD effective radius is not only noticeable in oceanic outflow regions but also extends to land regions. This is mainly due to the fact that different cloud regimes and thermodynamical conditions exist over the land and the oceanic regions in the selected study area. The covariability is especially stronger just off the eastern Chinese coast in the first two modes, suggesting a noticeable influence of pollution advection on the low-level clouds, possibly dominating over other factors that can also cause covariability.

### 4.3 MC analysis over BB1: Africa

The maximum covariance analysis applied to AOD and CD effective radius over this region reveals that only the first mode is the most significant and dominant mode explaining a variability of up to 20 %. The remaining modes do not exhibit statistically significant covariability and the SCFs in the subsequent modes are less than 10 %. Thus, here we focus only on the first mode. The covariability seen in this one mode only is nonetheless very significant and noteworthy, as this mode exhibits the regions where the Twomey effect is likely dominant, particularly over the pollutant outflow regions. The heterogeneous pattern in these two fields for this mode is shown in Fig. 10.

As mentioned earlier, this study region is dominated by the seasonal biomass burning in southern Africa that occurs during the months of June through November, transporting enormous amounts of absorbing aerosols across the southeast Atlantic over the extensive stratocumulus decks. This outflow region has therefore been the centre of many measurement campaigns and studies focusing on characterizing aerosols and clouds and their interactions (see, for example, Redemann et al., 2021). To what extent these aerosols affect the underlying stratocumulus decks is still the topic of intense research. Although the freshly emitted biomass burning aerosols are not ideal condensation nuclei, numerous studies have previously shown that the morphology, composition and size distribution can change considerably during the transport, making them one of the dominant local sources of condensation nuclei, thereby increasing cloud droplet number concentration and changing droplet size (Petters et al., 2009; Lu et al., 2018; Che et al., 2022; Royer et al., 2023).

The significant covariability between AOD and CD effective radius here indeed suggests that as these biomass burning aerosols are advected over the oceanic stratocumulus regions in the model simulation, they undergo ageing and mixing with other anthropogenic aerosols as well as ma-

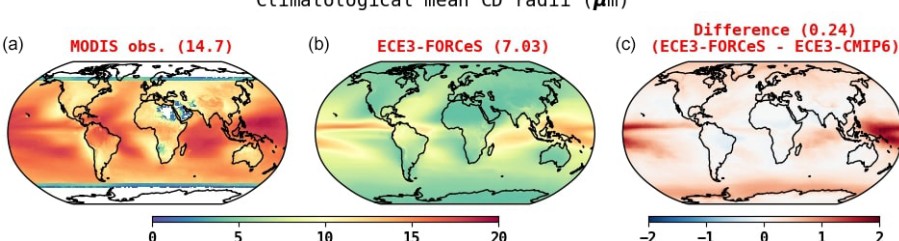

**Figure 6.** Climatological mean CD effective radius (µm) from MODIS observations **(a)** and ECE3-FORCeS simulations **(b)** and their differences with respect to ECE3-CMIP6 simulations **(c)**. The global mean values are given in brackets.

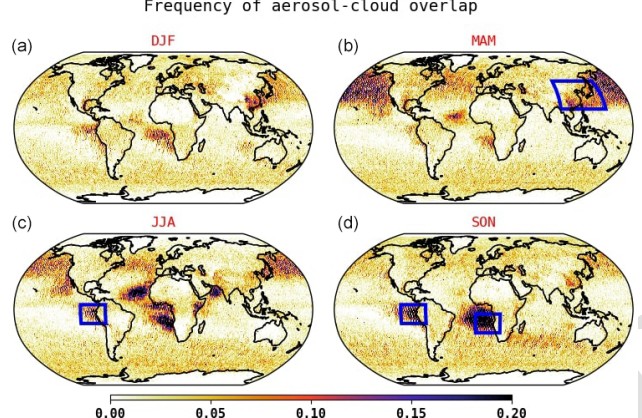

**Figure 7.** Frequency of aerosol–cloud overlap estimated based on 4 years (May 2006–June 2010) of CALIPSO satellite observations (Devasthale and Thomas, 2011). The regions selected are marked by the blue boxes.

rine aerosols, possibly becoming more efficient CCN. In the ECE3-AerChem model, the open biomass burning emissions are distributed up to 2 km following the vertical profiles for forest fires as given in Table A1 of van Noije et al. (2014). The accompanying advection of humidity, however, makes the aerosol–cloud interactions over these regions less predictable, as the local meteorology together with the changing aerosol ageing, coating and size distribution introduces high internal variability.

### 4.4   MC analysis over BB2: South America

Although the frequency of aerosol–cloud overlap is also high in this outflow region, the weakest covariability between AOD and CD effective radius is observed over the ocean compared with two previous study areas as shown in Fig. 11. Although the first mode alone accounts for 59 % of the total variability, the spatial extent of the statistically significant covariability and its magnitude is limited and weak over the ocean compared with the land. A number of factors that can affect the variability in AOD and clouds need to be considered here. The aerosols need to be convected

and advected over much longer distances in the free troposphere crossing the Andes mountain range (Bourgeois et al., 2015, 2018). The high residence time and ageing during the transport most likely lead to strong changes in aerosol properties and size distribution before they are available as condensation nuclei after the descent over the cooler upwelling waters off the western coast of South America. Furthermore, the large-scale dynamical variability associated with the El Niño–Southern Oscillation (ENSO) strongly influences the variability of low-level cloudiness in this region. ENSO also most likely induces variability in the background marine aerosols through changes in sea surface temperatures and winds. Therefore, the correspondence between AOD and CD effective radius at larger temporal scales is much weaker.

### 4.5   Implications for cloud radiative effects

Additionally, we examined the implications of this covariability between AOD and CD effective radius for the cloud radiative effects. To investigate this, a composite analysis is carried out based on the PCs of AOD, as shown in Fig. 12 as an example for mode 1 over eastern China.

Two composites are generated, one for which the PCs of AOD are greater than 0 and another where the PCs of AOD are less than 0. In the composite analysis for PCs of AOD > 0, we investigated the anomalies of TOA SW CRE, AOD, CD effective radius, CLWP, CDNC (cloud droplet number concentration), specific humidity and low cloud cover fraction. The composite analysis for the PCs of AOD < 0 is exactly the opposite of that obtained for PCs of AOD > 0. For brevity, we present the outcomes (Fig. 13) for the three modes specifically over eastern China for one composite analysis with PCs of AOD > 0.

In this composite, the anomalies in TOA SW CREs indicate pronounced cooling over the pollution source regions and the pollutant outflow region in modes 1 and 2, while the opposite is observed in mode 3, as can be seen in Fig. 13. The significant cooling observed in the first two modes can stem from two processes: either enhanced aerosol–cloud interactions (Twomey effect) and/or to an increase in low cloud cover, leading to brighter clouds and increased reflection. Regarding the first hypothesis, the AOD anomalies are in

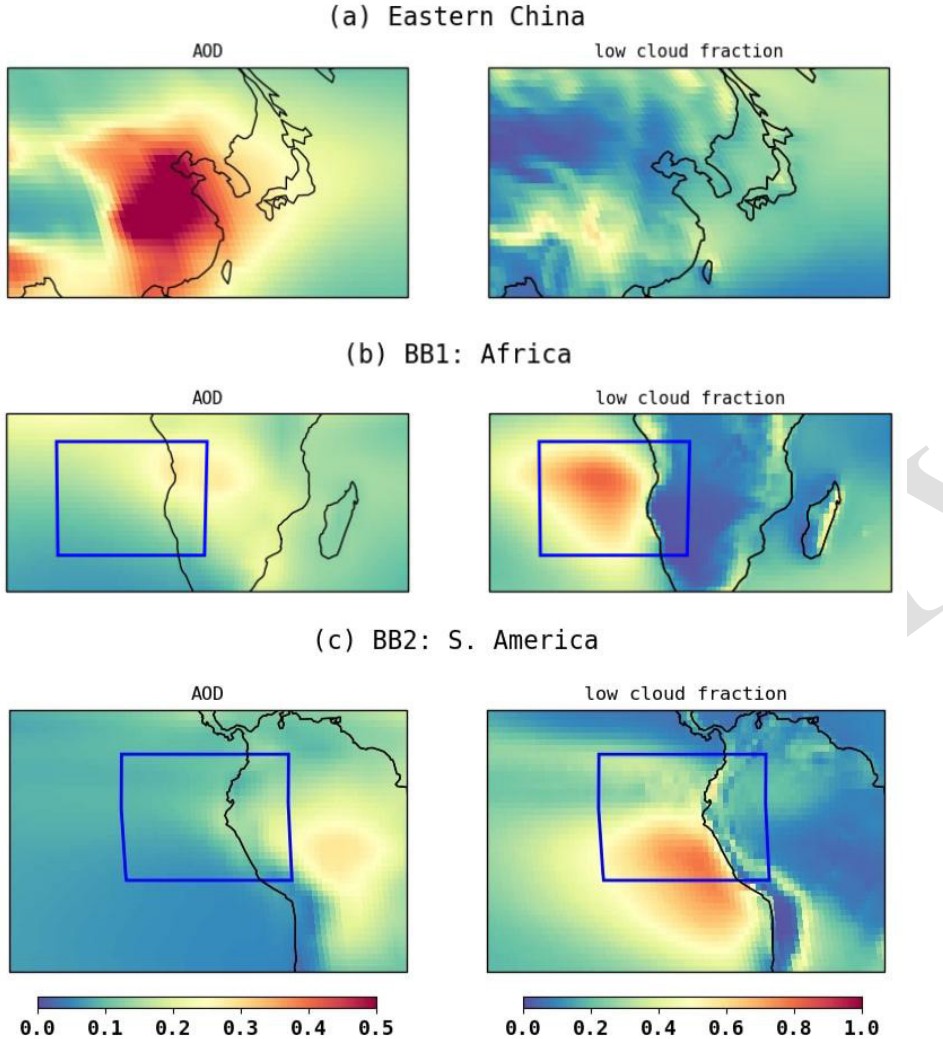

**Figure 8.** Climatological mean AOD (left) and low cloud fraction (right) over the selected three regions. **(a)** Eastern China. **(b)** BB1: Africa (blue box). **(c)** BB2: South America (blue box).

fact lower in this composite together with a corresponding positive anomaly in CD effective radius, indicative of larger droplets. The cloud droplet number concentration is also lower. This indicates that the Twomey effect cannot explain the cooling anomalies in the TOA SW CREs. However, a higher low cloud fraction in tandem with higher CLWP and specific humidity is seen. This higher low cloud fraction can adequately account for the marked cooling in TOA SW CREs. Consequently, the results show that, despite the existence of robust aerosol–cloud interactions at the larger spatio-temporal scales, the TOA SW CREs are predominantly driven by the changes in low-level cloud fraction in these simulations. Previous studies (Gryspeerdt et al., 2019) have shown the inverse relationship between CDNC and CLWP over the heavily polluted regions based on satellite observations. The results of this composite analysis are in line with those previous studies, suggesting that CDNC–

CLWP control can indeed have a greater impact on cloud radiative effects than the Twomey effect.

Similar conclusions can be derived for the first modes over the biomass burning regions of Africa and South America. The results are shown in the Appendix in Figs. A1 and A2 for Africa and A3 and A4 for South America.

## 5 Discussion and conclusions

In the framework of the EU project FORCeS, a number of significant improvements in the representation of aerosol and cloud processes were implemented in the ECE3-AerChem model, ECE3-FORCeS. ECE3-FORCeS now includes a representation of the atmospheric cycle of dust minerals that are relevant for their climate impacts as well as a new source of marine organic aerosols. As a result, the absorption of radiation by dust in the SW is now dependent on the on-

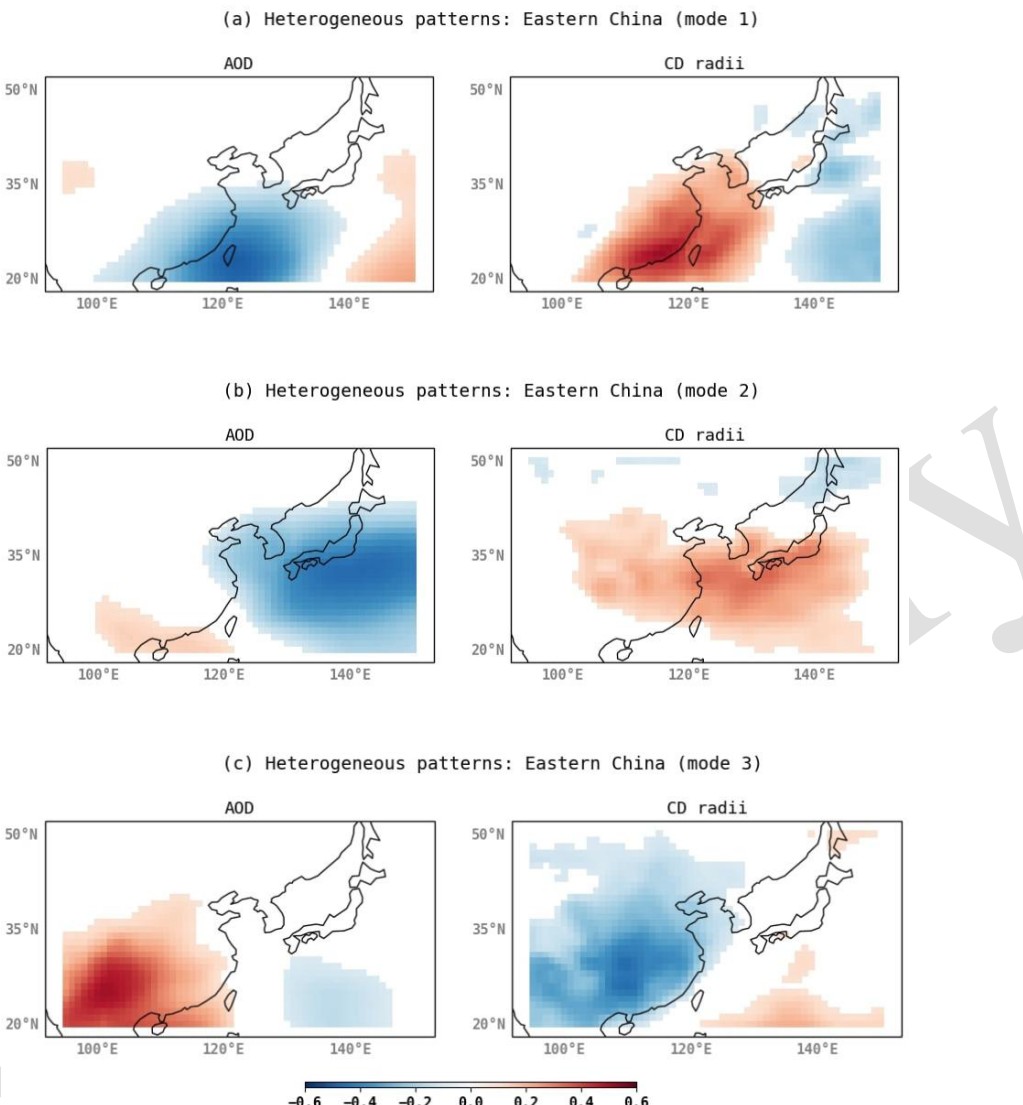

**Figure 9.** Statistically significant heterogeneous patterns over eastern China in the three modes **(a–c)** derived from the maximum covariance analysis.

line calculated abundance of hematite. The model also incorporates an aerosol-sensitive scheme for the estimation of primary ice crystals, which depends on the abundance of quartz, K-feldspars and marine organics for the immersion freezing and on dust and soot for deposition nucleation. This primary ice can lead to secondary ice production in supercooled conditions, which is quantified via RaFSIP (Georgakaki and Nenes, 2024). A new warm cloud activation scheme by Morales-Betancourt and Nenes (2014) is also implemented, enabling a more accurate estimate for the number concentration of droplets formed at the time of maximum supersaturation. Finally, a new thermodynamic module for inorganic aerosols, ISORROPIA-LITE, has also been implemented in the model.

These changes and improvements are bound to have an impact on the cloud properties and their radiative effects. Therefore, we used 41 years of historical atmosphere-only (AMIP) simulations for the period 1980–2020 from this recent FORCeS model version to confront them with the satellite-based observations. A suite of satellite sensor retrievals and simulators are used to facilitate the comparison. Due to the increased cloud fraction and cloud liquid water path compared with its previous CMIP6 version, the strong warm biases often seen over the Southern Ocean are reduced significantly in the FORCeS version of ECE3-AerChem. The bias in the SW TOA CREs is reduced by nearly 50 % over this region.

Using these simulations from the ECE3-FORCeS, we further carried out a maximum covariance analysis between

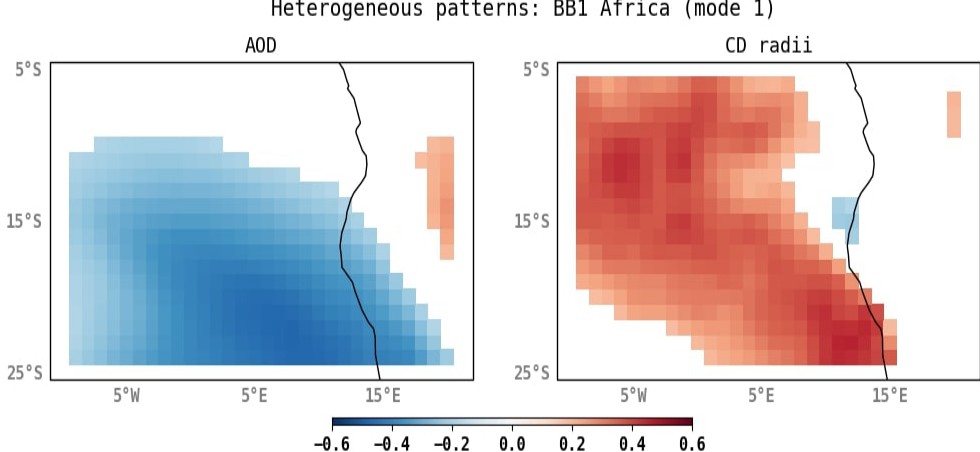

**Figure 10.** Statistically significant heterogeneous patterns in mode 1 derived from the MCA over a biomass burning (BB1) outflow region of Africa.

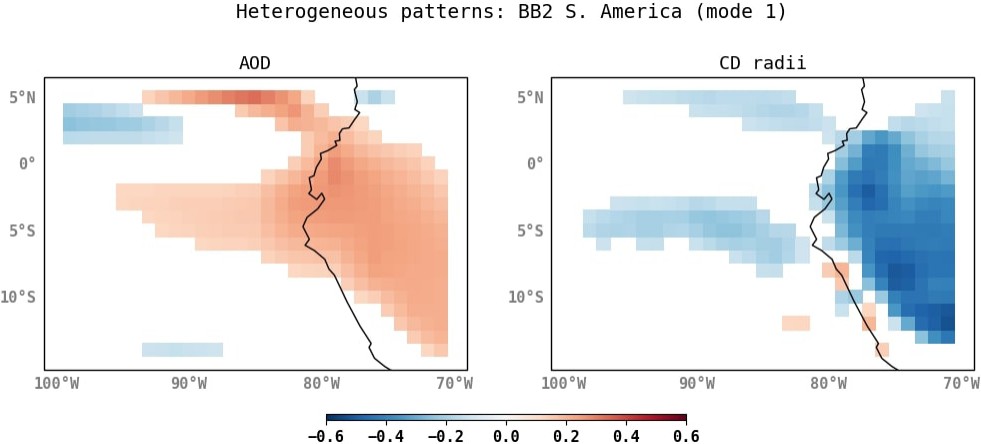

**Figure 11.** Statistically significant heterogeneous patterns in mode 1 derived from the MCA over a biomass burning (BB2) outflow region of South America.

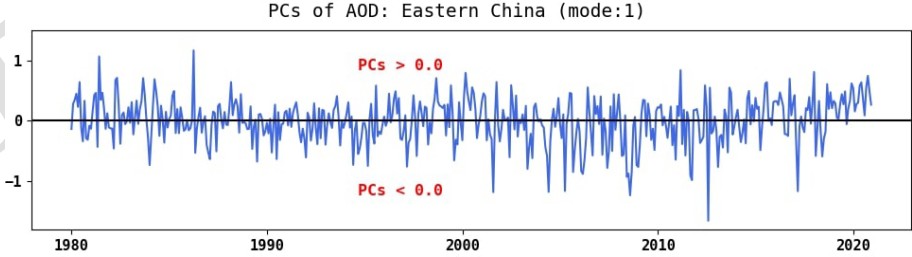

**Figure 12.** The monthly principal components of AOD corresponding to mode 1 of the MCA.

AOD and CD effective radius. The aim of this exercise was not to identify any new climate process but rather to test whether the covariability between AOD and CD effective radius could also be seen at the larger spatial and temporal scales. Among the three oceanic regions that were chosen based on the high frequency of aerosol and low-level cloud overlap, the strongest statistically significant inverse covari-

ability between AOD and CD effective radius was observed over the eastern Chinese outflow region. Here, the aerosol loading is generally quite high and the aerosol sources are mostly anthropogenic. The conditions favourable for facilitating aerosol–cloud interactions, such as persistence of westerly winds, high humidity and aerosol load in the lower troposphere, as well as favourable aerosol composition and

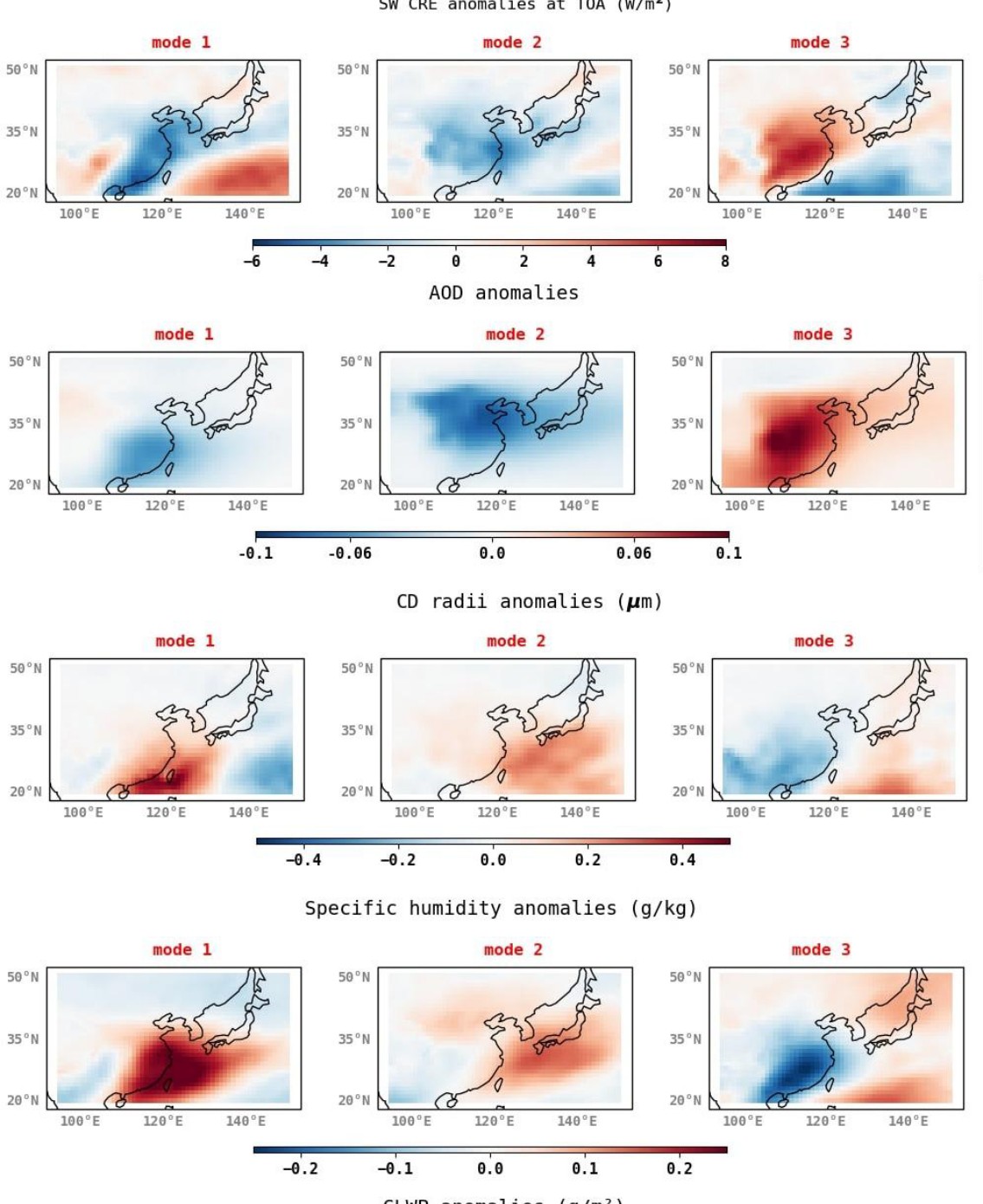

**Figure 13.**

size distribution over the outflow region likely lead to strong inverse covariability between AOD and CD effective radius seen even at larger spatio-temporal scales. This means that the Twomey effect can manifest itself even at a climate scale by dominating the other possible drivers of local variability. Note that the model simulations do not suffer from the limitations often discussed when using satellite data for similar analyses, such as seeing only cloud top information or sampling issues related to simultaneous aerosol and cloud retrievals. Furthermore, most of the aerosol outflow in this region occurs in the lowermost troposphere, and hence AOD

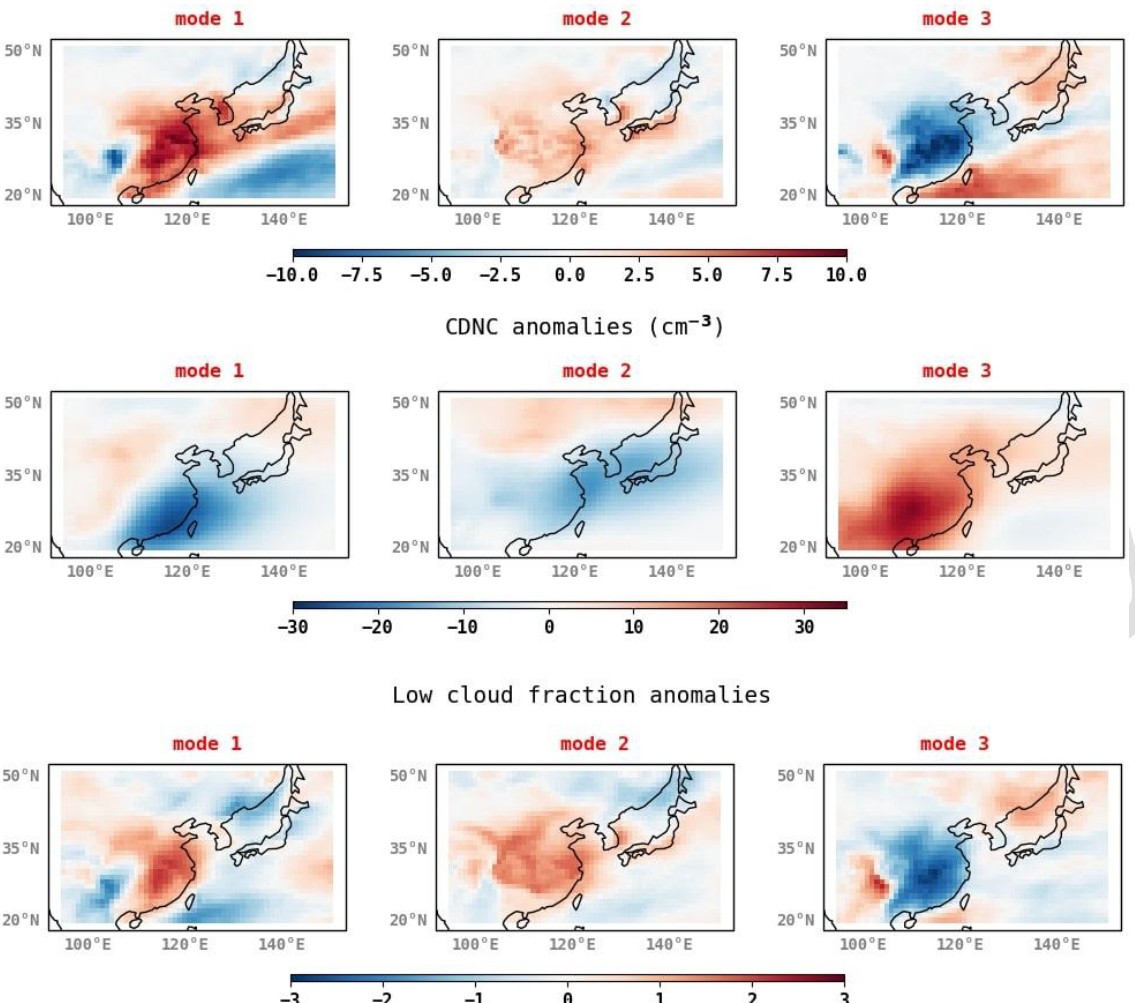

**Figure 13.** Each panel represents respectively the anomalies of TOA SW cloud radiative effects (W m$^{-2}$), AOD, CD effective radius (µm), specific humidity (g kg$^{-1}$), cloud liquid water path (g m$^{-2}$), CDNC (cm$^{-3}$) and low cloud fraction in the three modes over eastern China.

could indeed be a good proxy for the cloud condensation nuclei.

In the case of the other two regions in the southeast Atlantic (along the west coast of Africa: BB1) and eastern equatorial Pacific (along the west coast of South America: BB2), biomass burning is a significant source of aerosols during the months selected for the study. The covariance between AOD and CD effective radius is statistically significant only in the first mode of variability in these regions. However, it can be seen that the covariance is more pronounced over the pollutant outflow region in BB1 and the source region in BB2. This is most likely due to the fact that the aerosol vertical distribution and its variability are more disengaged from the variability in the underlying cloud decks due to longer transport distances in BB2, owing to the Andes mountain range and the subsequent changes in aerosol composition and size distribution that may not favour the covariability. In such case, the total AOD may indeed not be a proxy for

cloud condensation nuclei, as argued by the previous studies (Quaas et al., 2020). Our analysis also shows a strong cooling/warming in the TOA SW CREs despite strong covariability (larger droplets with low AOD and vice versa) and can be attributed to stronger/weaker control by the relationship between CDNC and CLWP. Further investigations are underway to carry out a similar study using satellite observations, but the lack of reliable, multidecadal data of AOD, CCN and CDNC is currently making such an analysis challenging.

The updates to the EC-Earth3-AerChem model described in this work improve the representation of aerosols and aerosol–cloud interactions. They address previously missing processes, such as secondary ice particles, and improve existing parameterizations, such as cloud droplet activation. These modifications make the model more realistic and closer to what is observed, but there are still biases in the cloud microphysical properties. One of the reasons may be that the model was re-tuned using a subset of the parame-

ters identified in the tuning strategy of the CMIP6 version of the model. However, finding a new set of tuning parameters to improve clouds while maintaining radiation balance, cloud forcing, surface temperatures, precipitation patterns, etc. is challenging. A complete re-tuning was beyond the scope of this project. Future model developments aim to reduce biases through new parameterizations for updraft velocity and secondary ice production (RaFSIP v2). The extent to which these changes, along with re-tuning, could mitigate the biases requires further investigation. The goal is to incorporate these improvements that were achieved during the FORCeS project into the next version of the EC-Earth model, which will then be used to contribute to CMIP7, particularly AerChemMIP, in order to provide a better understanding of the role of various aerosols in the climate and its sensitivity.

**Appendix A**

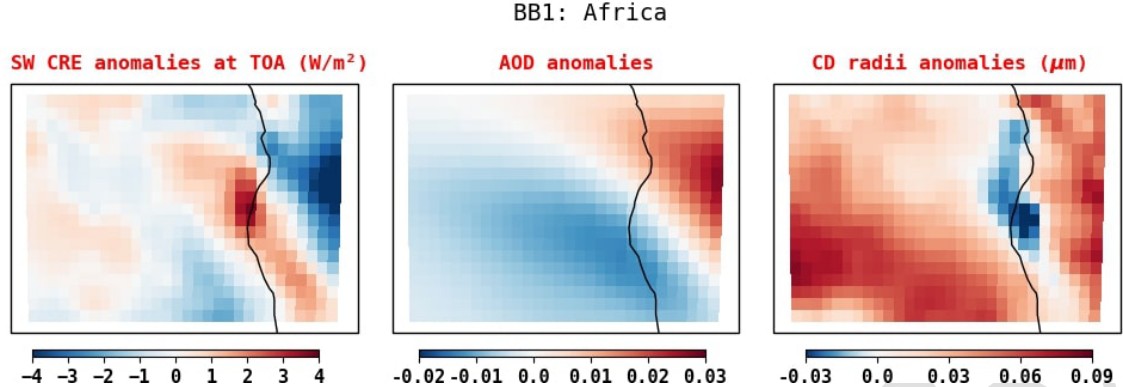

**Figure A1.** Anomalies of top-of-the-atmosphere SW CRE (W m$^{-2}$), AOD and CD radii (μm) in the three modes over BB1: Africa.

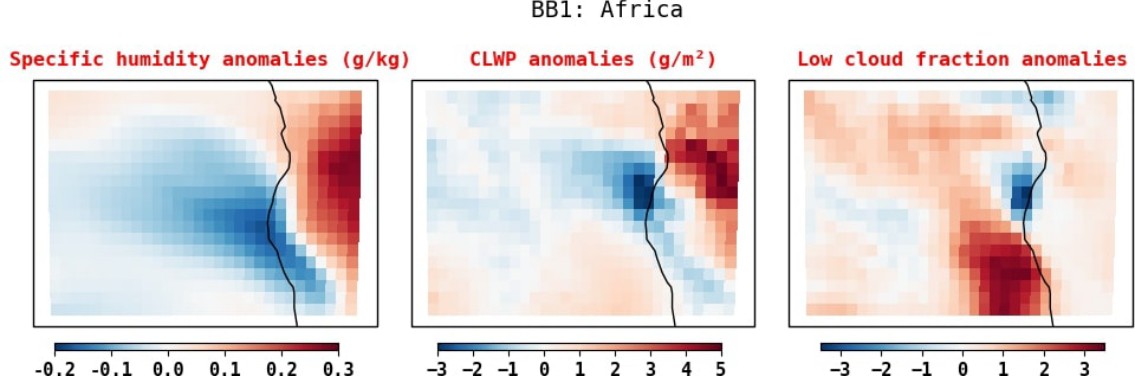

**Figure A2.** Anomalies of specific humidity (g kg$^{-1}$), CLWP (g m$^{-2}$) and low cloud fraction in the three modes over BB1: Africa.

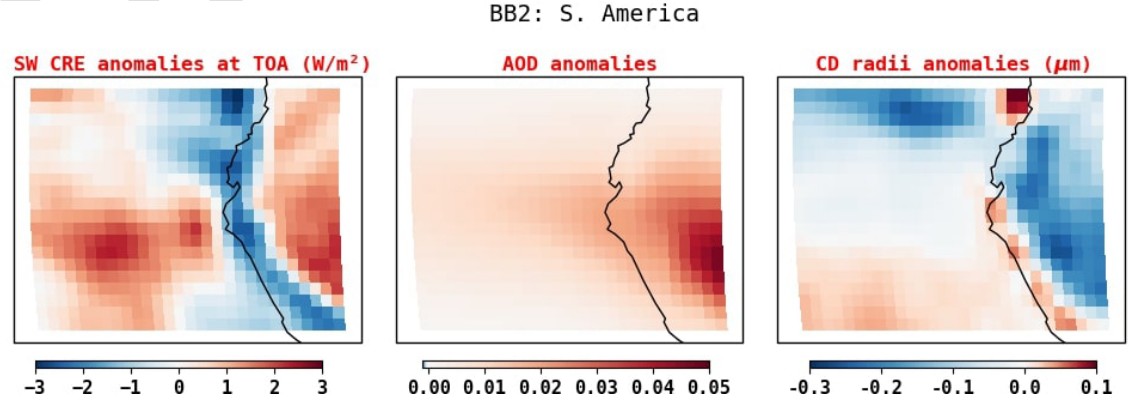

**Figure A3.** Anomalies of top-of-the-atmosphere SW CRE (W m$^{-2}$), AOD and CD radii (μm) in the three modes over BB2: S. America.

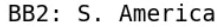

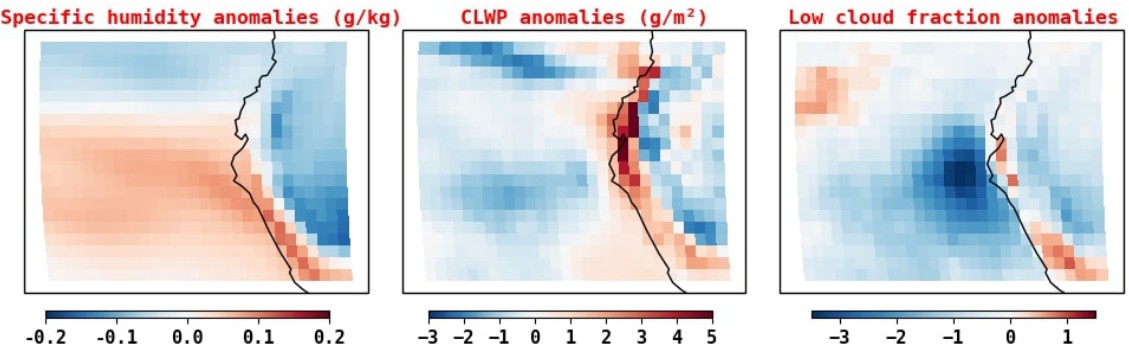

**Figure A4.** Anomalies of specific humidity ($\mathrm{g\,kg^{-1}}$), CLWP ($\mathrm{g\,m^{-2}}$) and low cloud fraction in the three modes over BB2: S. America.

*Code and data availability.* The EC-Earth3 code is available from the EC-Earth development portal for members of the consortium. All code related to CMIP6 forcing is implemented in the component models. The model codes developed at ECMWF, including the atmosphere model IFS, are the intellectual property of ECMWF and its member states. Permission to access the EC-Earth3 source code can be requested from the EC-Earth community via the EC-Earth website (http://www.ec-earth.org/about/contact/, last access: 2023, https://doi.org/10.26050/WDCC/AR6.C6ACEEEEA, EC-Earth-Consortium, 2023) and may be granted if a corresponding software license agreement is signed with ECMWF. The repository tag for the version of EC-Earth that is used in this work is "projects/FORCeS". Currently, only European users can be granted access due to license limitations of the atmosphere model. However, the model code is made accessible to both the editor and the reviewers throughout the review process. The component models NEMO, LPJ-GUESS, TM5 and PISM are not limited by their licenses.

All model output data are freely available from any ESGF data node as part of CMIP6 (https://esgf-data.dkrz.de/search/cmip6-dkrz/); search for `<model_id>=EC-Earth3-AerChem`, `<experiment_id>=amip` and `<variant_label>=r1i1p1f1` for ECE3-CMIP6 or `<variant_label>=r1i1p4f1` for ECE3-FORCeS. The Python scripts used to generate the figures in the paper, the Python code for the calculation of maximum covariance analysis and the observational data for model evaluation are archived in Zenodo under https://doi.org/10.5281/zenodo.10781927 (Thomas, 2024).

*Author contributions.* MAT and KW conceptualized the research. MAT performed the analysis and drafted the manuscript. SW post-processed the raw model data. AD provided and analysed the satellite data observations that were used in the analysis. MC, PG, MCS, MGA, MK, AN and CPGP were involved in the implementation of the different processes in the ECE3-AerChem model and contributed to the detailed text regarding these, as described in Sect. 3 of the paper. KW, TvN and PLS designed and carried out the simulations. All the co-authors contributed to the interpretation of the results and review of the paper.

*Competing interests.* The contact author has declared that none of the authors has any competing interests.

ther geographical representation in this paper. While Copernicus Publications makes every effort to include appropriate place names, the final responsibility lies with the authors.

*Acknowledgements.* The authors acknowledge the funding from the Horizon European Union 2020 Framework Programme (FORCeS project; grant no. 821205). Montserrat Costa-Surós has received funding from the European Union's Horizon 2020 research and innovation programme under the Marie Skłodowska-Curie (grant no. 754433) from H2020-MSCA-COFUND-2016 (STARS). Carlos Pérez García-Pando, Marios Chatziparaschos, Montserrat Costa-Surós and Maria Gonçalves Ageitos acknowledge the funding from the European Research Council under the H2020 Research and Innovation Programme through the Consolidator Grant, FRAGMENT (grant no. 773051); the AXA Research Fund through the AXA Chair on Sand and Dust storms at the Barcelona Supercomputing Center (BSC); and the support of the Department of Research and Universities of the Government of Catalonia to the Research Group Atmospheric Composition (code: 2021890 SGR01550). The model simulations and data storage were enabled by resources provided by the National Academic Infrastructure for Supercomputing in Sweden (NAISS) at Linköping University and have been partially funded by the Swedish Research Council (grant no. 2022-06725).

*Financial support.* This research has been supported by the European Horizon 2020 (grant no. 821205), the HORIZON EUROPE Marie Sklodowska-Curie Actions (grant no. 754433), the European Horizon 2020 (grant no. 773051), the AXA Research Fund (code: 2021890 SGR01550) and the Vetenskapsrådet (grant no. 2022-06725).

*Review statement.* This paper was edited by Po-Lun Ma and reviewed by two anonymous referees.

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

**Remarks from the typesetter**

TS1    Due to the requested changes in this paragraph, we have to forward your requests to the handling editor for approval. To explain the corrections needed to the editor, please send me the reason why these corrections are necessary. Please note that the status of your paper will be changed to "Post-review adjustments" until the editor has made their decision. We will keep you informed via email.