# Peer review of "Recent improvements and maximum covariance analysis of aerosol and cloud properties in the EC-Earth3-AerChem model"

_EGUsphere, 2024_

## Author Comment (AC1)

**Response to Anonymous Reviewer-1**

I think this manuscript is within the scope of GMD and does constitute a contribution to the field. The authors showcase (briefly) some recent improvements in an atmosphere model. They then proceed to applying some statistical analysis on the improved model.

The manuscript is decently written, but can be challenging to read. Most figures can be improved. The manuscript covers two distinct topics that at times come across as unrelated. Perhaps this could be two separate manuscripts focused on two separate topics? The model evaluation part is brief and incomplete; likewise, the maximum covariance part only touches the surface of the topic it is tackling.

I have written down some reactions, comments, and questions. I hope the authors find them helpful in revising their manuscript (if they choose to do so). I don't see any major issue in this manuscript and so I will be happy if it is published in the near future :)

We thank the reviewer for the encouraging comments and constructive suggestions. Please find below a point by point response to the queries raised.

The decision to consolidate the two topics into a single manuscript was based on the following:
1. The model evaluation section focuses exclusively on cloud properties, although the model improvements targeted both aerosols and clouds so as to reduce the uncertainties in estimating the radiative forcing exerted by aerosols.
2. The maximum covariance analysis technique employed in this study enables an in-depth exploration of the intricate interplay between aerosols and clouds. This analysis is particularly relevant in three distinct regions where low-level clouds and pollution outflows coincide and are prominent, providing valuable insights into their complex interactions.

By consolidating these elements into one manuscript, we aim to demonstrate how the recent improvements enable us to provide a comprehensive perspective on the interrelated nature of aerosols and clouds.

We have made an in-depth evaluation of all important cloud physical properties that we can obtain from satellites in the manuscript. We have further clarified this in Subsection 3 as 'Assessing the cloud properties with the improved ECE3-FORCeS model'. Though the maximum covariance analysis conducted in this study is at its present state self-contained, we agree that there is the possibility to explore more. We also primarily wanted to introduce and demonstrate the efficacy of this method in extracting information when two variables presumably covary.

**Lines 63--69: Not sure what this paragraph adds to the conversation here. Consider adding more context why this is needed or simply deleting.**

We agree with the reviewer as this information has already been given in the 'Acknowledgements'. This paragraph has been rewritten as follows ' Within the framework of the FORCES project (https://forces-project.eu/), this study aims to identify and implement the crucial aerosol and cloud processes that could potentially reduce uncertainties in aerosol forcing estimates. This is detailed in Section 2.2. These improvements are being incorporated into the three global models that were previously used for CMIP6 (https://pcmdi.llnl.gov/CMIP6/): NorESM2-LM (Seland et al., 2020), MPI-ESM1.2-HAM (Mauritsen et al., 2019; Neubauer et al., 2019) and EC-Earth3-AerChem (van Noije et al., 2021). This paper focuses on the simulations carried out with the EC-Earth3-AerChem model.'

**Lines 76--77: This may warrant further explanation. As a somewhat informed reader, this left me wondering: What's the effective resolution of this model in sum? At what resolution are things like macrophysics and microphysics schemes are done? It is fine to refer to citations; segueing into this with a "however" left me wondering about several questions. So, I would either consider expanding slightly here (e.g., why you felt the need to point out this specific piece of information) or simply delete to avoid reading getting stuck like I was.**

The chemistry and aerosol module, TM5, is computationally expensive and is run at a much coarser resolution compared to the atmospheric model. The cloud macrophysics and microphysics are part of the atmosphere model and so are run at the same T255L91 resolution. These resolution details are typically provided when describing the model configuration to facilitate tracking of the specific resolutions employed for a particular evaluation or analysis. This has now been clarified in the revised manuscript.

**Line 84: Why brief? Isn't this manuscript supposed to detail these updates?**

Here, by 'brief', we meant that we will not be detailing the impacts of all individual improvements (Section 2.2.1 - Section 2.2.5) with respect to the CMIP6 model version. This will be published separately. Here, we are focussing mainly on evaluating the resulting improvements in cloud properties after these updates.

**Lines 85--94: Since the Patridge et al. manuscript is in preparation, I worry this entire paragraph (section) lacks substantiation. On the one hand, someone like me having read the studies cited a few times in the past can nod in agreement about the assertion that the 2014 scheme is likely better. On the other hand, how can you convince others that your assertions are sound besides taking your words for it? A potential avenue is weakening the assertions and citing counterarguments (e.g., the meta-modeling study in doi:10.1175/JAS-D-15-0223.1 can be interpreted as saying these models perform similarly an older comparison study doi:10.1029/2011MS000074 show differences between schemes).**

We agree with the reviewer and this section has been revised as follows:

[revised manuscript text omitted]

**Line 122: Perhaps use a more conventional notation for the "approximately bigger than" unless you mean something different.**

The sentence now reads as 'In regions such as the remote Arctic/Antarctica where the primary ice nucleating particles (INP) are sparse (Moore et al., 2024; Wex et al., 2019), a considerable source of ice crystals is via secondary ice production (SIP) processes in supercooled clouds *(temperatures higher than approximately -10ºC)* (Järvinen et al., 2022; Järvinen et al., 2023).

**Lines 134--136: Any reasoning for this choice to parameterize this process?**

The notion of the Ice Enhancement Factor (IEF) is frequently employed in the literature to indicate the prevalence of secondary ice production (SIP). This is a promising approach for parameterizing the effects of SIP in mixed-phase clouds because it allows direct comparison with the IEF (ICNCs/INPs) derived from in-situ observations or retrieved from remote-sensing data (e.g., Wieder et al., 2022). Unlike conventional SIP parameterizations (e.g., Phillips et al., 2017), the RaFSIP scheme requires only a few input variables, offering a streamlined and easily implementable solution for large-scale models without detailed microphysics schemes, which would not otherwise support the explicit treatment of SIP. Section 2.2.4 is revised accordingly.

**Lines 145--148: These tunings may require some more detail and clarification. For example, I have no idea what RPRCON is. And second, the last sentence comes out of nowhere. Third, I have no idea what you meant by cloud forcing --- you mean cloud radiative effects like you refer to them later or something different? Altogether, the section left me wondering: What did the other preceding changes do to the forcing? Is the ~1 W/m2 solely from these three tunings or from everything together?**

As mentioned in the Section 2.2.6, tuning is done to the model after major updates to minimize the radiative imbalance at the TOA and at the surface (global mean net radiative flux at the TOA, TOA and surface SW and LW flux, cloud radiative effects in LW and SW) in

AMIP simulations with CERES-EBAF data set as reference. There is a set of tuning parameters associated with each model and in this study, only three of these parameters were adjusted. They were 1. RPRCON, which is, as mentioned in the paragraph, the conversion efficiency, i.e. the rate of autoconversion of cloud water to rain, 2. the critical radius for autoconversion and 3. the standard deviation of the updraft velocity distribution in the cloud droplet activation.

The combined model updates led to a TOA net radiation imbalance of more than -3 W/m2 with the old settings of the tuning parameters. In contrast, the CMIP6 version of the model had an imbalance of -0.46 W/m2. By adjusting the above parameters we were able to bring the model closer to a radiative balance and a net TOA imbalance of -0.7 W/m2.

Section 2.2.6 is revised to incorporate the above mentioned details.

**Line 181: What is meant by rotated here?**

The different modes in MCA are obtained through singular value decomposition (SVD) of the covariance matrix between the two input fields. These modes are orthogonal and explain the maximum covariance between the fields. But, the resulting patterns can be complex and difficult to interpret physically. However, in the 'Rotated' MCA, the orthogonality constraint is relaxed to obtain easier to interpret patterns in terms of physical processes. In this case, the patterns are simplified by maximizing the variance across each mode. In our study, Varimax rotation technique was used. This has been clarified in the revised text.

**Line 189: What is meant by an invariant quantity here?**

This means that, in this case, the squared covariance fraction remains constant even if the modes are modified or transformed in some way. This has been added to the revised text.

**Lines 190--214: I do not necessarily see this as needed in the main manuscript, but if the authors deem it necessary, I would advise to reorganize the section. I would start with the raw data X, Y (as an aside, what do a x n and b x n actually mean?) and then develop Ax and Ay using the SVD before defining the correlations. I would also add --- in equation form --- what "Corr" means. Finally, I would not have hom, het, Corr, and SCF be italicized.**

This section has been rewritten based on the reviewer's suggestions.

**Lines 216--220: Is this AMIP run completely compliant with published protocols? If not, state deviations. It would be good to simply state the full description of forcers and settings. Also, 1980--2020 AMIP vs 1980--2018 CMIP6?**

Both AMIP simulations are fully compliant with the CMIP6 AMIP protocol, but have been extended beyond the CMIP6 historical period using anthropogenic and biomass burning emissions of ozone and aerosol precursors, GHG concentrations and other forcings following the CMIP6 protocol for the SSP2-4.5 scenario experiment. SSTs and sea ice fractions are prescribed using the CMIP6 version (1.1.8) of the PCMDI forcing data set, which extends out through 2021. We have clarified this in the text.

**Figure 1 and associated text: In the bottom panel in the figure, please consider having the color bar go from -0.25 to +0.25 (i.e., symmetric). Also, any guess which processes (line 240) have led to the changes you're describing?**

We understand the thinking behind having a symmetric colorbar, but we have left the figure as it is as a symmetric colorbar will not highlight the differences that are less than 10% as shown in the figure below:

[Figure]

We have additionally added a sub plot showing the differences in the simulated cloud fraction with respect to CALIPSO observations as suggested by the second reviewer. With respect to ECE3-CMIP6 simulations, an increase in cloudiness is simulated in the high latitude regions. This increase mainly stems from the increase in low level cloud fraction. And this can be attributed to the model updates (for eg. the cloud droplet activation scheme) that targets liquid water clouds. This is mentioned in the revised manuscript.

**Figure 2 and associated text: Consider having the same y-axis for all (maybe 0.0 to 1.0) and consider adding axis labels.**

Following the review suggestion, the range of the y-axis is kept the same for all the subplots and the axis labels have been added to the figure. This is updated in the revised manuscript.

**Figure 3 and associated text: Maybe I am more optimistic than you, the model seems to be doing okay for stratocumulus regions. Maybe show the spatial distribution of the difference if you want to highlight the difference?**

Thank you for the suggestion. The difference plot and a description are added to the revised manuscript. The difference plot captures more than what meets the eye and it is a good idea to show the difference plots.

Here, we can actually see that the total cloud fraction is underestimated by the model over the tropics and the mid latitudes by 20-25%, however, the water cloud fraction is slightly overestimated by 10% and the model simulates well the ice cloud fraction over these regions compared to the MODIS observations.

**Figure 4 and associated text: I would have appreciated seeing the total means on these figures like you showed earlier. Again, consider having the difference plot with a symmetric color bar (-70 to +70 or so). Given the plots as well as the discussion, maybe it is worth showing the combined (SW + LW) CRE?**

The global means are shown in the revised figure. Here, again a symmetric color bar will not highlight the negative CRE differences as clearly as it is presented now. The combined (SW+LW) plot would show a spatial pattern more similar to the SW-only plot and also, to reduce the number of subplots, this will not be included in the revised manuscript. We have used a different colorbar so that the white regions correspond to 'no changes'.

[Figure]

**(b) Differences (ECE3-CMIP6 - CERES-EBAF obs.)**

SW (2.07) LW (-0.96)

**(c) Differences (ECE3-FORCeS - CERES-EBAF obs.)**

SW (0.23) LW (0.55)

**Figure 5 and associated text: As you say, these plots should be remade with better color bars. Also, because of the significant differences (underestimation by half), it may be worthwhile to give more reasoning and context. How worried should you be about this weakness of the model? Are there any tunings or strategies to remedy the situation? Have other modeling centers faced similar problems?**

The figures have been revised with one colorbar for both CLWP and CIWP in the subplots. The updated model version has reduced the CLWP bias but increased the CIWP bias. Better model tuning could potentially improve cloud water path, both liquid and ice. We essentially re-tuned the model using a subset of the parameters identified in the tuning strategy of the CMIP6 version of the model, and we have clarified the tuning procedure in the main text. However, finding a new set of tuning parameters to improve clouds while maintaining radiation balance, cloud forcing, surface temperatures, precipitation patterns, etc., is challenging. A complete re-tuning was beyond the scope of this project and paper.. Future model developments aim to reduce biases through new parametrizations for updraft velocity and secondary ice production (RaFSIP v2). The extent to which these changes, along with re-tuning, could mitigate the biases requires further investigation. Model improvement is an ongoing process, and more effort is needed to further reduce biases.

**Figure 6 and associated text: What's going on with MODIS near the poles? If no values are available there, maybe it is worth denoting that with white (like in Figure 7) as opposed to 0. Also, consider extending the color bar to 0.**

The colorbar is extended to zero and the regions where MODIS observations are lacking are set to white. The minimum value of cloud droplet radius is 4 microns in the models.

**Figure 7 and associated text: Oh, so this comparison kind of makes no sense. Why not add cloud-top diagnostics to your model? The MODIS simulator can do that or a simple cloud-top algorithm can do the job too. I would remove this comparison if there is no better comparison. There is no need to muddy the water with badly constructed comparison here in my opinion.**

We agree with the reviewer here. We have removed this comparison as we do not have the corresponding COSP output for fairer CDNC evaluation.

*Figure 9 and associated text: I think it would be better to redo the figure and* **ensure the color bars are the same for all AOD panels on the one hand, and all cloud fraction panels (ideally, 0 to 1) on the other hand.**
The figures are replotted incorporating the reviewers suggestions.

**Line 393 and thereabouts: You're essentially describing the aerosol effective radiative forcing, but somehow only discussing two of the well known mechanisms. An example of a missing one is precipitation suppression. How did you exclude it from your two hypothesized processes? I am glad you're citing the Gryspeerdt et al. 2019 paper, but I am not sure your text gives a proper summary of the convoluted nature of all of this...**
The precipitation suppression is indeed an important process. Here, we were thinking mainly from the top of the atmosphere perspective of cloud radiative effects. The changes in cloud microphysics and cloud cover have the largest impact on the changes in TOA CREs. Precipitation suppression, although not directly affecting TOA CREs, does impact the cloud lifetime and cloud cover. However, here we see that in the composite where PCs of AOD > 0 the absolute AOD is lower than the other AOD. We therefore suspect that the increase in cloud fraction is probably not linked to the increase in precipitation suppression.

**Figure 14: Is the figure spreading on two pages intentionally? Maybe reorganize it so that it fits on one page?**
The figures are intentionally spread on to two pages to ensure clarity.

**The entirety of section 4: Please say more about the data sampling and processing. How many years were included? When the data was put in the statistical analysis framework, what was the frequency of sampling or was it just monthly/yearly means? What are the implications? From Figure 13, it appears you're using 40 years worth of data, monthly averaged? Maybe this was somewhere in the text but I missed?**
Please note that the Section 4 has been reorganized as follows:
4 Results-II: Maximum Covariance Analysis: Covariability between AOD and CD effective radius
4.1 Regions selected for the analysis
4.2 MC analysis

Maximum covariance analysis is applied to AOD and CD radii to assess the covariability between these two variables. Since the analysis focuses on the months when the frequency of aerosol and cloud overlap is higher in the outflow regions, as explained in Section 4.1, the temporal sampling differs regionally. For instance, as shown in Table-1, we use all 492 months (41 years) of data for Eastern China. However, over the biomass burning regions of Africa and South America, the temporal sampling is limited to 122 and 246 months,

respectively. Since the focus of MCA analysis has been on testing the visibility of Twomey effect at a climate scale, the monthly means are analyzed. Before applying MCA, both the variables are deseasonalized and detrended based on the respective data for each region so as to ensure that this analysis captures the covariability between the two variables unbiased by the seasonal or long term trends.

The heterogeneous patterns, the patterns that bring out the areas of covariability will be presented for the three regions selected for the study. Only the regions exhibiting statistical significance at the 95% significance level are showcased. The following subsections detail the first three modes of the analysis over the regions selected for this study.

**The entirety of section 4: Maybe putting lat/lon ticks (and their values) on the figures will be helpful?**
Fig. 9 is replotted to clearly show the selected regions. Lat/lon ticks have been added to all the figures in section 4 in the revised manuscript.

**The entirety of section 4: Maybe doing the same analysis on both model versions (the improved and the one before it) will be interesting?**
We did actually plan to do this, but decided not to include such a comparative analysis in the end mainly for pragmatic reasons. First, we already did a thorough evaluation of cloud properties between those two versions of EC Earth. And second, we believe it will not add more value to the present analysis since the results suggest that AOD-CDradii co-variability and Cloud cover - CRE relationship will likely dominate in both model versions. In future, we do however plan to perform a comparative analysis with the satellite based observations.

**Line 434 and thereabouts: Would you mind elaborating regarding what you mean here? For example, I think studying things at cloud top is likely better than studying them elsewhere; at the end of the day, that's where the radiation is reflected, no?**
Yes, indeed the changes at the cloud top are important here from the top of the atmosphere perspective. We simply wanted to remind the readers that the typical limitations associated with the analysis of satellite based retrievals from the passive imagers do not apply in our case and that the co-variability signal we see is indeed physical.

**(contd) Another, how do your model runs avoid the sampling issues related to simultaneous aerosol and cloud retrievals? How are you analyzing/sampling your model data?**
In the model evaluation section, for comparisons with satellite data, we used COSP-derived variables whenever those were available so as to avoid issues related to sensitivity of satellite based instruments and time sampling. Furthermore, it is to be noted that, unlike the passive

satellite imagers that provide cloud information at the top and the total column aerosol measure such as AOD, the model estimates are available for each atmospheric layer.

**Lines 437--438: Say more. Why could it be a good proxy? What makes you say so?**
AOD has been used as a good proxy for the following reasons:

1. The Eastern China region selected for the co-variability study is the region where the pollution outflow just off the eastern coast is predominantly concentrated in the lower tropospheric levels.

2. Not all aerosols can serve as cloud condensation nuclei; for example, over the outflow regions over Eastern China, the aerosols are of more anthropogenic in origin and can be more efficient CCNs compared to the biomass burning aerosols over the other two regions.

Also, CCN was not one of the standard variables of EC-Earth3 output.

**Lines 449: Based on what specifically can it be attributed? What you showed in the maximum covariance analysis? I am not sure if that's as clear cut as you're saying here, so I encourage further discussion.**
This is explained in Section 4.5 in the original manuscript. This is supposed to be a separate paragraph and we have rephrased this sentence and summarized a bit more details as given below.
'We also further analyzed the implications for TOA SW CREs associated with this covariability. This was evaluated based on the PCs of AOD.'

**Code and data: Thank you for making your analysis scripts readily available! Maybe mention that you're also including the MC code out there?**
A sentence is added to the revised manuscript mentioning this.

---

## Author Comment (AC2)

**Response to Anonymous Reviewer-2**

This study illustrated the importance of aerosols, clouds, and their interactions in the climate system and the potential impact of accurately modeling these processes on the uncertainty of future climate projections, and analyzed in detail the latest improvements in aerosol and cloud properties and maximum covariance analyses in the EC-Earth3-AerChem model. This study is of great significance for improving the accuracy of global Earth system models in climate prediction. It is recommended that the manuscript can be published after minor revisions.

We thank the reviewer for the encouraging comments and constructive suggestions. Please find below a point by point response to the queries raised.

**1. In studying the covariance of AOD and CD, why was the maximum covariance analysis method used? What are the advantages over other analysis methods?**

MCA is an efficient method which identifies coupled patterns that explain the maximum covariance between the two variables. Similar tools, such as, for example, canonical correlation analysis (CCA) aim to find patterns with maximum temporal correlation, which may not necessarily explain much covariance. MCA is simpler to implement and interpret, and is robust compared to other methods like CCA and regression. This is clarified in the revised manuscript.

**2. The spatial distribution of the difference between ECE3-FORCeS and observations can be added in Fig. 1 to reflect the comparison between simulations and observations.**

We have added this subplot in Fig. 1.

**3. The article mentions in section 3.1 that the ECE3-FORCeS model better reproduces the spatial distribution of the total cloud amount, but is biased higher in the polar regions and explains that it is due to low clouds. Could you explain more about the bias. What caused the bias to be much higher in the polar regions than in the equatorial and mid-latitude regions?**

**4. It looks to me that the changes in cloud fraction are mostly at high latitude regions. Does that mean the updates within FORCeS only work for limited regions?**

We are addressing 3 & 4 here:

The FORCeS project aimed at improving the representation of aerosol and cloud processes in Earth System Models. Some of the model updates target liquid water clouds (e.g. for EC-Earth3 the cloud droplet activation) while other improvements were done for ice clouds (e.g. secondary ice production). Bringing it all together in one model could lead to trade-offs between the different developments that possibly could explain why model performance

hasn't uniformly improved. Model development is a continuous process, future versions of the EC-Earth model will address remaining biases.

**5. Line 380: Please add relevant references.**

The following two references are added to the revised manuscript.

Bourgeois, Q., A. M. L. Ekman, and R. Krejci (2015), Aerosol transport over the Andes from the Amazon Basin to the remote Pacific Ocean: A multiyear CALIOP assessment, J. Geophys. Res. Atmos., 120, 8411–8425, doi:10.1002/2015JD023254.

Bourgeois, Q., Ekman, A. M. L., Renard, J.-B., Krejci, R., Devasthale, A., Bender, F. A.-M., Riipinen, I., Berthet, G., and Tackett, J. L.: How much of the global aerosol optical depth is found in the boundary layer and free troposphere?, Atmos. Chem. Phys., 18, 7709–7720, https://doi.org/10.5194/acp-18-7709-2018, 2018.

**6. It is recommended that the conclusion section further explicitly summarize the contribution of model improvements to climate prediction and the innovation and limitation of this study.**

The following paragraph is added to the manuscript 'The updates to the EC-Earth3-AerChem model described in this work improve the representation of aerosols and aerosol-cloud interactions. They address previously missing processes, such as secondary ice particles, and improve existing parameterizations, such as cloud droplet activation. These modifications make the model more realistic and closer to what is observed, but, there are still biases in the cloud microphysical properties. One of the reasons may be that the model was re-tuned using a subset of the parameters identified in the tuning strategy of the CMIP6 version of the model. However, finding a new set of tuning parameters to improve clouds while maintaining radiation balance, cloud forcing, surface temperatures, precipitation patterns, etc., is challenging. A complete re-tuning was beyond the scope of this project. Future model developments aim to reduce biases through new parametrizations for updraft velocity and secondary ice production (RaFSIP v2). The extent to which these changes, along with re-tuning, could mitigate the biases requires further investigation. The goal is to incorporate these improvements that were achieved during the FORCeS project, into the next version of the EC-Earth model which will then be used to contribute to CMIP7, particularly AerChemMIP, to provide a better understanding of the role of various aerosols in the climate and its sensitivity.'